# Hotspots of human mutation point to clonal expansions in spermatogonia

Vladimir Seplyarskiy[1,2,4 ✉], Mikhail A. Moldovan[1,4 ✉], Evan Koch[1], Prathitha Kar[1], Matthew D. C. Neville[3], Raheleh Rahbari[3] & Shamil Sunyaev[1,2 ✉]

In renewing tissues, mutations conferring selective advantage may result in clonal expansions[1–4]. In contrast to somatic tissues, mutations driving clonal expansions in spermatogonia (CES) are also transmitted to the next generation. This results in an effective increase of de novo mutation rate for CES drivers[5–8]. CES was originally discovered through extreme recurrence of de novo mutations causing Apert syndrome[5]. Here, we develop a systematic approach to discover CES drivers as hotspots of human de novo mutation. Our analysis of 54,715 trios ascertained for rare conditions[9–13], 6,065 control trios[12,14–19] and population variation from 807,162 mostly healthy individuals[20] identifies genes manifesting rates of de novo mutations inconsistent with plausible models of disease ascertainment. We propose 23 genes hypermutable at loss-of-function (LoF) sites as candidate CES drivers. An extra 17 genes feature hypermutable missense mutations at individual positions, suggesting CES acting through gain of function. CES increases the average mutation rate roughly 17-fold for LoF genes in both control trios and sperm and roughly 500-fold for pooled gain-of-function sites in sperm[21]. Positive selection in the male germline elevates the prevalence of genetic disorders and increases polymorphism levels, masking the effect of negative selection in human populations. Despite the excess of mutations in disease cohorts for 19 LoF CES driver candidates, only 9 show clear evidence of disease causality[22], suggesting that CES may lead to false-positive disease associations.

The genome of the average newborn harbours roughly 70 de novo point mutations, of which roughly 80% arise in paternal germline[14,23]. De novo mutation is a common cause of sporadic monogenic disease[9–13,15], which has led to a lot of international effort collecting large cohorts of parent–child trios[9–13,15]. These trios typically include unaffected parents and their children who, depending on the cohort, may be affected by conditions ranging from neurodevelopmental disorders (NDD) to congenital heart diseases[9–13,15]. To discover the genes associated with disease, these studies compare the observed counts of de novo mutations in a gene with the ones expected from a baseline mutation rate model[9,12]. Owing to the initial ascertainment on disease phenotype, this signal provides evidence for the gene's causal role.

Studies of de novo variation, most prominently studies of sporadic NDD[9] and autism spectrum disorders (ASD)[12,15], have identified hundreds of disease genes. As expected, most were found to be under strong negative selection consistent with a major impact of de novo mutations on severe early-onset conditions[9]. However, we observe that a small subset of the identified genes defy a simple dependence of disease effect and selection and show substantial levels of population polymorphism. Another observed discrepancy is that counts of specific high-impact de novo mutations in disease trio cohorts exceed the maximal levels explainable by disease ascertainment. A potential explanation is that forces beyond random mutagenesis increase the observed incidence of mutations.

Many genes within this subset are involved in carcinogenesis or clonal expansions in somatic tissues suggesting that a plausible explanation may be offered by the positive selection leading to clonal expansions in spermatogonia (CES). Clonal expansions have been recently found to be a feature of all renewing tissues[1–4], including spermatogonia. Indeed, gain-of-function (GoF) mutations in 13 genes[7,8] have already been demonstrated to drive CES. In contrast to somatic tissues, CES increases the likelihood of transmitting driver variants to the next generation, elevating the observed mutation rate in the offspring (Fig. 1a). For this reason, the first CES drivers were discovered through their recurrence in patients with monogenic disease[5,6]. By increasing mutation rate, CES can lead to an increased prevalence of disease. Conversely, genes that drive CES without directly causing disease could appear as disease-causing due to the elevated number of de novo mutations observed in affected children.

To systematically investigate the possible effect of CES on human germline mutation and identify potential CES driver candidates, we developed a statistical approach leveraging human de novo mutation data[9–15,23] along with population genetic variation resources[20]. Most of the identified genes follow biological expectation for drivers of clonal

[1]Department of Biomedical Informatics, Harvard Medical School, Boston, MA, USA. [2]Brigham and Women's Hospital, Division of Genetics, Harvard Medical School, Boston, MA, USA. [3]Cancer, Ageing and Somatic Mutation, Wellcome Sanger Institute, Hinxton, UK. [4]These authors contributed equally: Vladimir Seplyarskiy, Mikhail A. Moldovan. ✉e-mail: alicodendrochit@gmail.com; Mikhail_Moldovan@hms.harvard.edu; ssunyaev@hms.harvard.edu

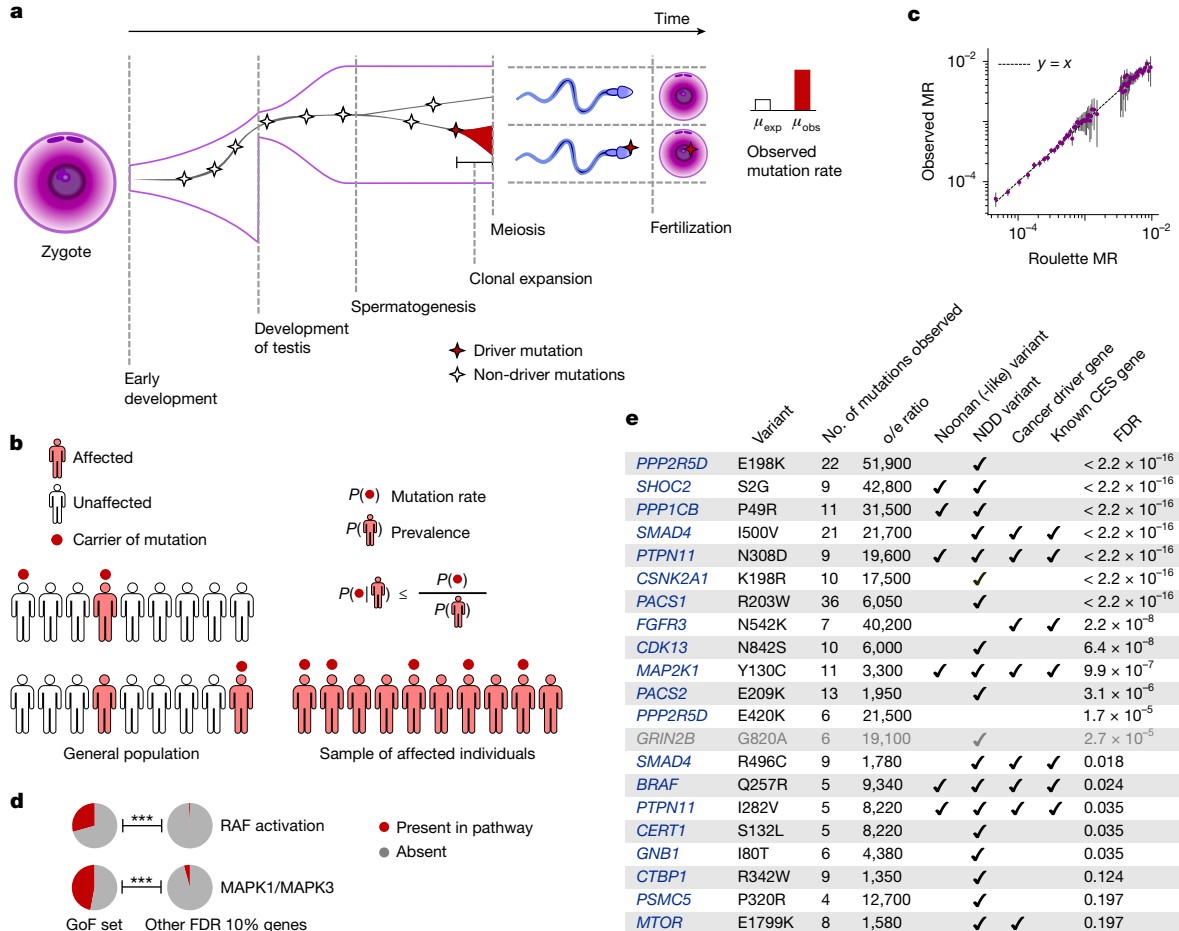

**Fig. 1 | CES and disease ascertainment. a**, Effect of clonal expansions in sperm on the observed mutation rate. **b**, In a randomly sampled cohort of individuals affected by a specific disease, the frequency of any variant is elevated due to ascertainment by at most the inverse of disease prevalence. **c**, Observed synonymous de novo variant counts in the NDD cohort published in ref. 9 stratified by Roulette mutation rate (MR) bins versus Roulette predictions. 95% Poisson confidence intervals are shown. **d**, Fractions of genes included in Reactome RAF activation and MAPK1/MAPK3 pathways. $P$ values of Fisher's exact test for comparison of fractions below 0.001 are shown as triple asterisks (***). **e**, CES driver candidates due to GoF. o/e ratio, observed-to-expected ratio.

| Variant | | No. of mutations observed | o/e ratio | Noonan (-like) variant | NDD variant | Cancer driver gene | Known CES gene | FDR |
|---|---|---|---|---|---|---|---|---|
| *PPP2R5D* | E198K | 22 | 51,900 | | ✓ | | | < 2.2 × 10⁻¹⁶ |
| *SHOC2* | S2G | 9 | 42,800 | ✓ | ✓ | | | < 2.2 × 10⁻¹⁶ |
| *PPP1CB* | P49R | 11 | 31,500 | ✓ | ✓ | | | < 2.2 × 10⁻¹⁶ |
| *SMAD4* | I500V | 21 | 21,700 | | ✓ | ✓ | ✓ | < 2.2 × 10⁻¹⁶ |
| *PTPN11* | N308D | 9 | 19,600 | ✓ | ✓ | ✓ | ✓ | < 2.2 × 10⁻¹⁶ |
| *CSNK2A1* | K198R | 10 | 17,500 | | ✓ | | | < 2.2 × 10⁻¹⁶ |
| *PACS1* | R203W | 36 | 6,050 | | ✓ | | | < 2.2 × 10⁻¹⁶ |
| *FGFR3* | N542K | 7 | 40,200 | | | ✓ | ✓ | 2.2 × 10⁻⁸ |
| *CDK13* | N842S | 10 | 6,000 | | ✓ | | | 6.4 × 10⁻⁸ |
| *MAP2K1* | Y130C | 11 | 3,300 | ✓ | ✓ | ✓ | ✓ | 9.9 × 10⁻⁷ |
| *PACS2* | E209K | 13 | 1,950 | | ✓ | | | 3.1 × 10⁻⁶ |
| *PPP2R5D* | E420K | 6 | 21,500 | | ✓ | | | 1.7 × 10⁻⁵ |
| *GRIN2B* | G820A | 6 | 19,100 | | ✓ | | | 2.7 × 10⁻⁵ |
| *SMAD4* | R496C | 9 | 1,780 | | ✓ | ✓ | ✓ | 0.018 |
| *BRAF* | Q257R | 5 | 9,340 | ✓ | ✓ | ✓ | ✓ | 0.024 |
| *PTPN11* | I282V | 5 | 8,220 | ✓ | ✓ | ✓ | ✓ | 0.035 |
| *CERT1* | S132L | 5 | 8,220 | | ✓ | | | 0.035 |
| *GNB1* | I80T | 6 | 4,380 | | ✓ | | | 0.035 |
| *CTBP1* | R342W | 9 | 1,350 | | ✓ | | | 0.124 |
| *PSMC5* | P320R | 4 | 12,700 | | ✓ | | | 0.197 |
| *MTOR* | E1799K | 8 | 1,580 | | ✓ | ✓ | | 0.197 |

expansions such as involvement in MAPK[7] and other major signalling pathways. We validated our findings using the already discovered CES genes[7,8], cohorts of unaffected trios[12,14–19] and comparison with the results of the direct sperm sequencing[21].

Rates of DNA damage, together with imperfections of DNA repair and replication, determine the baseline de novo mutation rate in the male germline[24]. Positive selection in spermatogonia elevates observed mutation rates of the drivers relative to the baseline (Fig. 1a). The approaches to identify CES genes taken here will generally compare observed mutation counts with those expected under a model of baseline mutation rate.

Historically, mutation rates at functional sites have been estimated from the prevalence of phenotypes caused by mutations at those sites. We generalized this logic to identify CES genes in a large trio sequencing cohort ascertained by any phenotype. Specifically, we counted de novo mutations in the largest assembled NDD trio cohort[9] (31,058 affected probands). The effect of ascertainment has a strict upper bound given by the inverse prevalence of ascertained phenotype. Denoting $V$ the presence of a de novo variant and $D$ the presence of the disease, we have:

$$\frac{\text{No. of obs}}{\text{No. of exp}} \approx \frac{P(V|D)}{P(V)} = \frac{P(D|V)}{P(D)} \leq \frac{1}{P(D)}, \quad (1)$$

where $P(V)$ is the probability of de novo variants defined by a mutation rate model (Supplementary Text 1) and $P(D)$ is the overall disease prevalence (Fig. 1b). Unlike disease-causing genes with no CES effect,

CES drivers could violate this upper bound, offering an approach for finding CES drivers. We develop a statistical approach based on equation (1) and demonstrate that it is robust to non-monogenic inheritance (Supplementary Text 4).

From the biological standpoint, because many known CES driver mutations cause NDD[7,8], we may expect the uncharacterized CES drivers to be similarly prone to causing NDD. The count of CES-driving mutations in the NDD cohort should be greater than in an unascertained set of trios, maximizing the chance of identifying CES drivers involved in NDD. Genes that drive CES but do not cause NDD can be found by this approach only if the effect of CES significantly exceeds the maximal effect of disease ascertainment given by the inverse prevalence (Supplementary Text 4).

This approach is sensitive to misspecifications of the mutation rate model. For the mutation rate model Roulette[25], we demonstrated the accuracy of predictions for the synonymous variants in the NDD cohort (Fig. 1c, Methods section 'Control for biases in Roulette' and Supplementary Tables 1–4).

Precision of the prevalence estimates, $P(D)$, might also affect the applicability of equation (1). To address this issue, we adopted a conservative lower bound on prevalence[26] of NDD of 1% (Methods section 'Phenotypic homogeneity in the NDD cohort'). We also show that the phenotypic sampling is uniform across the NDD subcohorts ruling out false-positive CES findings due to heterogeneous ascertainment by phenotype (Methods section 'Prevalence of NDD').

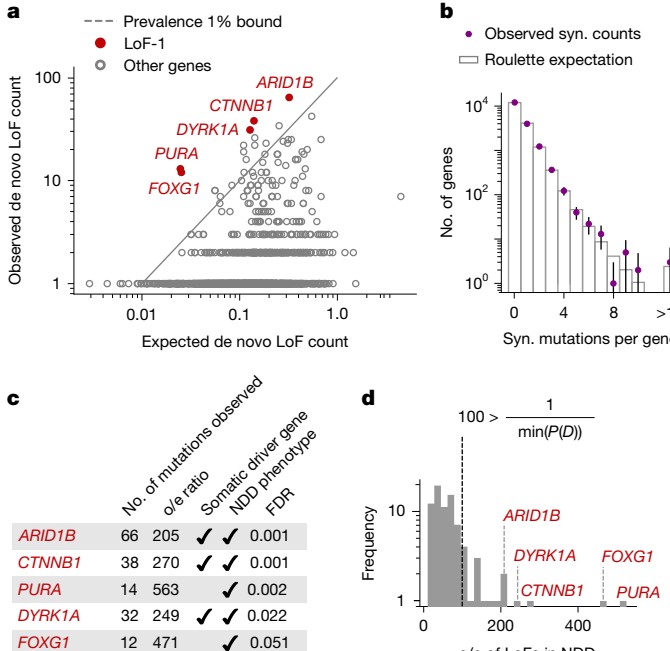

**a**
Observed de novo LoF count

- - - Prevalence 1% bound
● LoF-1
○ Other genes

*ARID1B*
*CTNNB1*
*DYRK1A*
*PURA*
*FOXG1*

Expected de novo LoF count

**b**
No. of genes

● Observed syn. counts
▢ Roulette expectation

Syn. mutations per gene

**c**

| | No. of mutations observed | o/e ratio | Somatic driver gene | NDD phenotype | FDR |
|---|---|---|---|---|---|
| *ARID1B* | 66 | 205 | ✓ | ✓ | 0.001 |
| *CTNNB1* | 38 | 270 | ✓ | ✓ | 0.001 |
| *PURA* | 14 | 563 | | ✓ | 0.002 |
| *DYRK1A* | 32 | 249 | ✓ | ✓ | 0.022 |
| *FOXG1* | 12 | 471 | | ✓ | 0.051 |

**d**

$$100 > \frac{1}{\min(P(D))}$$

Frequency

*ARID1B*
*DYRK1A*  *FOXG1*
*CTNNB1*  *PURA*

o/e of LoFs in NDD

**Fig. 2 | LoF-1 set of putative CES drivers. a**, Observed versus expected LoF counts in the NDD cohort. LoF-1 genes are shown in red. **b**, Observed synonymous (syn.) de novo mutation counts in the NDD cohort versus the ones expected under the Poisson counts around Roulette estimates. 95% Poisson confidence intervals are shown. **c**, Properties of LoF-1 genes. **d**, Ratio of observed-to-expected de novo LoF variant counts for genes at FDR < 0.1 in the NDD cohort published in ref. 9. The outliers with the ratio above the maximal ascertainment (LoF-1 genes) are highlighted in red.

## GoF mutations causing CES

All CES genes discovered to date act through GoF mechanisms, and all known CES mutations are individual missense variants[7,8]. We therefore first apply equation (1) to all possible ($5 \times 10^7$) individual missense de novo mutations in the NDD cohort and identify 21 variants in 18 unique genes that pass the 20% false discovery rate (FDR) threshold (Extended Data Fig. 1a and Supplementary Table 11). Among the 18 identified genes, 5 have been previously experimentally established to cause CES[7,8]. From the functional perspective, the identified genes are enriched in RAF activation (adjusted $P = 1.7 \times 10^{-7}$) and MAPK1/MAPK3 (Fig. 1d, adjusted $P = 2.2 \times 10^{-7}$) pathways[27] in agreement with previous studies[7] (Supplementary Table 12 and Methods section 'Expression and gene set enrichment analyses'). All but one of these genes are expressed in spermatogonia ($P = 6.5 \times 10^{-7}$). The exception, *GRIN2B*, was thus excluded from the GoF set (Extended Data Fig. 2) as a potential false positive.

As explained above, the test based on equation (1) has greater power to identify CES drivers that also cause NDD. Indeed, 16 out of 17 genes have independent evidence of association with disease phenotypes through GoF variants (Fig. 1e).

## LoF mutations causing CES

Although existing experimental work has focused solely on GoF mutations driving CES, loss-of-function (LoF) mutations are known to also drive clonal expansions in cancers and healthy renewing tissues[1]. This motivated us to investigate LoF mutations as plausible CES drivers (Fig. 2a). Because every gene harbours many possible LoF mutations, we may extend equation (1) to test sets of variants within genes rather than individual variants. We validated that the aggregation of LoF sites by gene does not bias our mutation rate expectation (Fig. 2b, Methods section 'Control for biases in Roulette' and Supplementary Table 5).

Five genes show numbers of de novo LoF mutations in the NDD cohort significantly exceeding any plausible ascertainment by disease (by equation (1)): *PURA*, *ARID1B*, *CTNNB1*, *DYRK1A* and *FOXG1* (Fig. 2c,d and Extended Data Fig. 1b,c). We call this list the LoF-1 set. Three of them, *ARID1B*, *CTNNB1* and *DYRK1A*, are involved in carcinogenesis or other clonal expansions[28,29] (Fig. 2c). Although *PURA* and *FOXG1* have not been previously identified as drivers of somatic clonal expansions, their functions fit the profile of clonal expansion drivers. *PURA* is involved in replication and transcription control and *FOXG1* is a transcription factor regulating early development (Fig. 2c).

The independent evidence of association with disease phenotypes supports the effect of LoF mutations on NDD for all five of these genes. In addition, these genes are highly selectively constrained in the human population (upper bound estimate of the observed-to-expected ratio of LoF single-nucleotide variant (SNV) counts (LOEUF) less than 0.3), consistent with their role in severe paediatric conditions.

## CES and LoF polymorphism

We now bring our attention to CES drivers with weak or no effect on NDD. The approach outlined above is not suited well to identify such genes, because it requires the counts of de novo mutations in the NDD cohort to exceed the expectation by at least 100-fold (inverse prevalence of NDD). Therefore, we rely on extra considerations. In particular, we note that CES should increase the rate of functionally consequential mutations and thus polymorphism levels in the general population. The rate of de novo mutations in a disease cohort should be elevated for both CES genes and disease genes. However, unlike CES driver with no effect on NDD, disease-causing genes should be deprived of functional polymorphism in the predominantly healthy population. This discrepancy between amounts of genetic variation in the population among genes significant for the excess of de novo variants in the NDD cohort over the mutation rate expectation (not necessarily exceeding the boundary of equation (1)) yields a procedure for identifying CES genes. Specifically, a gene with both large amounts of polymorphism in the general population and the excess of de novo variants in a disease cohort may be deemed a CES driver candidate. Note that this procedure may be applied to cohorts with mixed disease phenotypes, because the expectation is defined just by the mutation rate model and is independent of ascertainment, motivating us to increase power by merging de novo variation in cohorts ascertained by NDD (31,058 affected probands)[9] and ASD (16,877 affected probands)[12]. We quantify LoF polymorphism in the general population using LOEUF[20].

In total, 122 genes show a significant excess of LoF de novo mutations in the ASD–NDD cohort at FDR < 0.1 (Methods section 'Construction of the LoF-2 set' and Supplementary Tables 7 and 14). From these, we selected 19 genes (Fig. 3a and Extended Data Figs. 3 and 4) with LOEUF > 0.5 (concordant with cut-offs usually chosen for relaxed negative selection[12,30–32]). High values of LOEUF may be indicative of both relaxed selection and misspecifications of the mutation rate model, as should be the case with CES-driven mutation rate elevation. Genes with misannotated LoF variants[33,34] or involved in clonal expansions in blood[3] may also have spuriously high LOEUF values without CES. For these reasons, four genes were flagged as potential false findings (Supplementary Text 2 and 3). The remaining 15 genes we call the LoF-2 set.

Four extra considerations support this set of genes as CES driver candidates. First, we show that LoF-2 genes have a significant excess of de novo LoF mutations not just in the joint ASD–NDD dataset, but also in a healthy trio cohort[12,14–19] (Fig. 3b and Supplementary Tables 9 and 10). The excess of mutations in the healthy trio cohort could not be affected by disease ascertainment, allowing us to estimate the average CES effect for LoF-2 genes. The magnitude of the effect in LoF-2 genes is comparable between ASD–NDD and control cohorts (Fig. 3b, 18.7 versus 26.9, $P = 0.2$), indicating that the effect of disease ascertainment is most

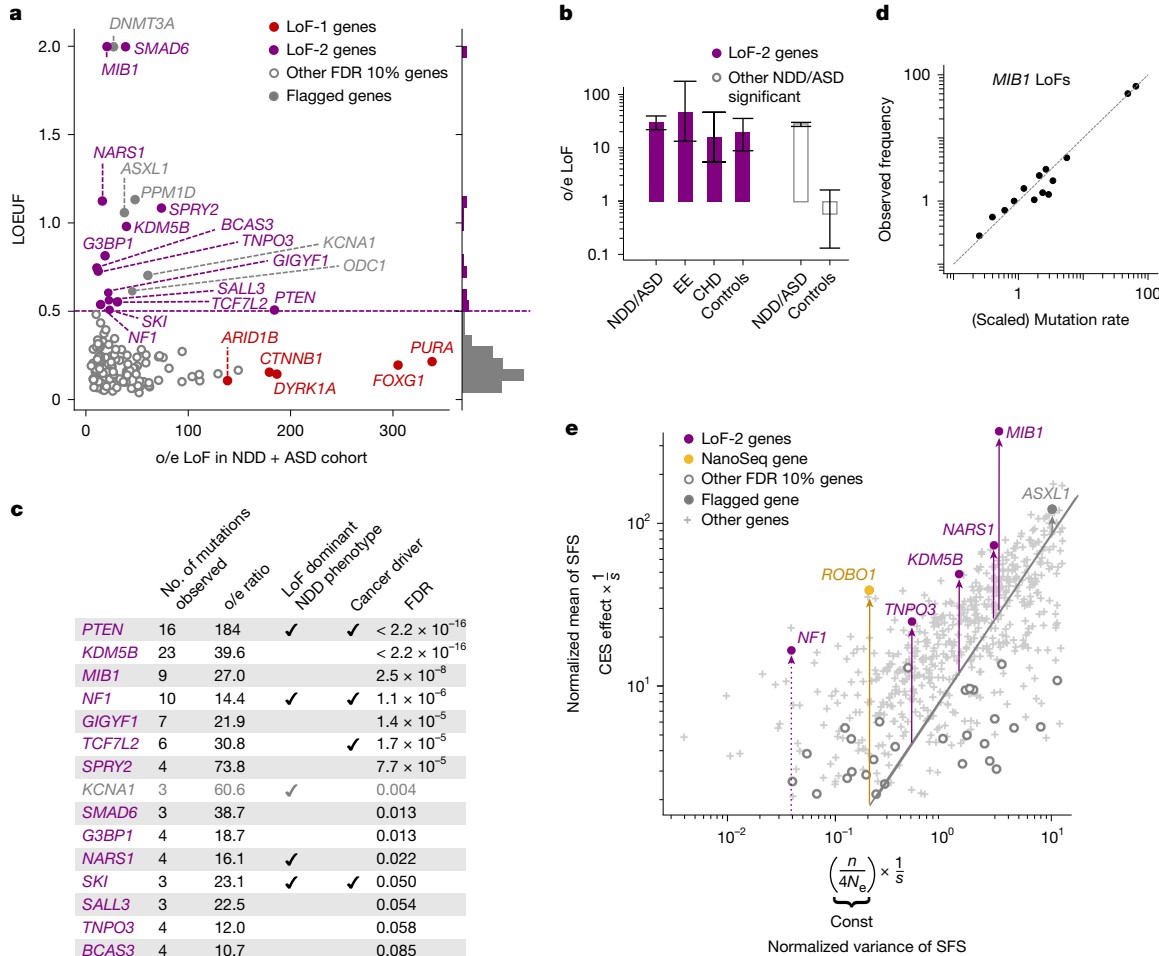

**Fig. 3 | Effect of CES on de novo mutations in disease and on the levels of LoF polymorphism in population. a**, Observed-to-expected variant count ratio for de novo LoFs in genes with FDR < 0.1 in the NDD cohort[9] merged with the ASD[12] cohort plotted against LOEUF scores. The horizontal dashed violet line indicates the 0.5 threshold for LOEUF used to construct the LoF-2 set, and the histogram to the right shows LOEUF values. **b**, Observed-to-expected LoF variant count ratio in four different trio cohorts. Violet bars indicate the observed-to-expected variant count ratios of LoF-2 genes and empty grey bars indicate other genes significant at FDR < 0.1 in the ASD–NDD cohort. Whiskers indicate the 95% Poisson confidence intervals. EE, encephalopathic epilepsy;

CHD, congenital heart disease. **c**, Properties of LoF-2 genes. The single gene not expressed in spermatogonia (*KCNA1*) is shown in grey. **d**, Frequency of LoF in gnomAD-v4 against mean scaled mutation rate predicted by Roulette in *MIB1*. LoF sites were aggregated by mutation rate and each point represents the mean taken for at least ten sites. **e**, The *x* axis shows inflation of the variance of the LoF allele frequencies in gnomAD-v4 due to random genetic drift proportional to the inverse selection coefficient (1/*s*). The *y* axis shows mean allele frequency scaled by mutation rate equal to CES effect multiplied by 1/*s*. Each point corresponds to an individual gene with at least ten LoF CpG sites.

probably small and most LoF-2 genes do not cause ASD or NDD. Indeed, only four genes from this set have independent evidence of the causal role in ASD–NDD[22] (Fig. 3c). Just one of these genes, *PTEN*, demonstrates an excess of de novo LoF mutations in the NDD cohort comparable to the excesses in the LoF-1 set (Fig. 3a,c). In line with small effects of NDD ascertainment in LoF-2 genes, we observe similar magnitudes of the enrichment of de novo LoF mutations observed in other trio cohorts ascertained by different phenotypes: encephalopathic epilepsy and congenital heart disease. Finally, using the cohort of healthy controls, we also estimated that just 15 LoF-2 genes explain 3.8% of the entire de novo LoF variation genome-wide.

To investigate the robustness of the LOEUF threshold used to select genes in the LoF-2 set, we compared the excesses of LoF mutations in the control cohort for different thresholds. The magnitude of the effect does not change substantially in the range of LOEUF threshold values from 0.3 to 2.0 (Extended Data Fig. 4). The effect in the healthy cohort dissipates for ASD–NDD genes with LOEUF < 0.3, consistent with causal disease effect of these genes.

Second, all LoF-2 genes barring *KCNA1* are expressed in spermatogonia (expression enrichment *P* = 0.01, Mann–Whitney *U*-test). *KCNA1*

was flagged as a potential false-positive finding (Extended Data Fig. 2). Four LoF-2 genes are also known to cause cancers[28].

Third, with the help of population variation data, we can find further evidence that the high level of LoF polymorphism in LoF-2 genes probably reflects elevated de novo mutation rate rather than relaxed selection. The large sample size of gnomAD-v4 (1.6 million haploid genomes) enables analysis of frequencies of LoF variants generated by recurrent mutations[25,35–37]. LoF mutations within the same gene are expected to have identical functional effects, including on CES. All individual LoF sites within the same gene therefore share a common linear inflation factor relative to baseline mutation rate expectations. In the presence of CES, allele frequencies remain linearly proportional to the baseline mutation rate attesting to CES being a linear effect, as illustrated by the example of *MIB1* in Fig. 3d (Supplementary Table 19).

We expand this logic to statistically detect elevation of mutation rate for LoF-2 set in the human population data. Variants with the same mutation rate and selection coefficient have different allele frequencies due to the effect of genetic drift. Strong selection overpowers the effect of drift and makes allele frequencies approach the expectation under the balance between mutation and selection. CES is expected

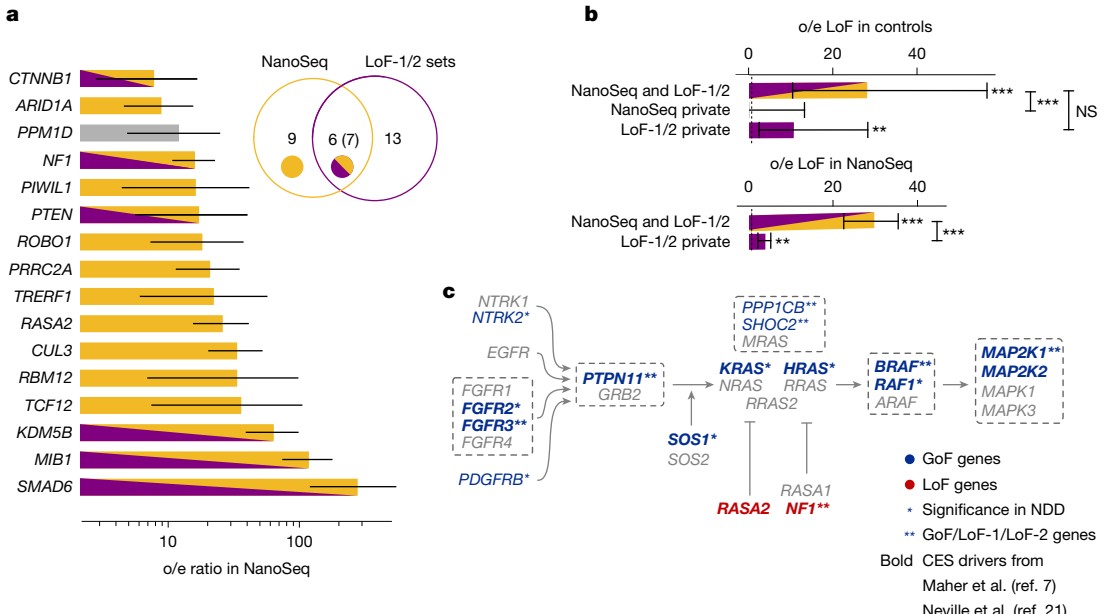

**Fig. 4 | Comparison with direct sequencing data. a**, Observed-to-expected ratio for LoFs counts in NanoSeq sequencing of sperm for genes with significant excess of LoFs. Poisson-conjugate (gamma) 95% confidence intervals are shown. Grey bar indicates that *PPM1D* has been flagged in our previous analysis. Venn diagram shows the numbers of genes in sets 1 and 2 of genes significant due to LoFs in our study, number of genes significant due to LoFs in the NanoSeq study[21] and the number of common genes. Numbers in parentheses include *PPM1D*. **b**, Enrichments of de novo mutations in controls for genes overlapping between NanoSeq and LoF-1 + LoF-2, genes private to NanoSeq and genes private for set 1 and set 2. The number of asterisks indicates the binomial significance threshold: *0.05, **0.01, ***0.001; NS, not significant. **c**, Core pathway of RAF activation. Grey dashed frame shows protein complexes, activator and repressor activities are denoted with pointed and blunt arrows, respectively.

to shift this balance and increase the average allele frequency, but is not expected to affect deviations from this balance. Slightly more formally, the distribution of sampling allele frequencies in the presence of recurrent mutation and relatively strong selection, is given by the Nei approximation[38] (Methods section 'Segregation of LoF-2 polymorphism in the human population'). Relying on the Nei approximation, we isolate variance inflation due to genetic drift. It is inversely proportional to selection coefficient. The mean allele frequency is similarly inversely proportional to selection coefficient, but is also directly proportional to the effect of CES. Thus, CES effects may be identified from the comparison of the mean and variance terms for a single gene. Even with the current sample sizes, this procedure has low power, as most of the information is contained in frequencies of highly mutable yet rare CpG sites, and so we restricted our analysis to 952 genes with more than 10 CpGs. Figure 3e shows the evidence of CES effects for LoF-2 genes included in the analysis.

Our final analysis relies on GeneBayes[39], which evaluates per-gene selective constraint relying on a protein function-informed prior and a likelihood based on LoF variation within the gene. The prior, unlike the likelihood, is not influenced by mutation rate and the effects of CES. We observe LoF-2 genes have the most inconsistent prior and likelihood estimates, concordant with the effects of CES on the likelihood (Extended Data Fig. 5 and Supplementary Table 8).

## Comparison with sperm sequencing

Direct sequencing of mature sperm provides important data for the evaluation of CES effects without any ascertainment by phenotype. In parallel with this study, NanoSeq, an accurate single-molecule resolution DNA sequencing technology, was applied to sequencing sperm cells of 63 donors[21]. A panel of 263 genes implicated in cancer was sequenced with a mean coverage of $8 \times 10^4$ and whole exomes were sequenced with a mean coverage of $2.1 \times 10^4$. This study identified 40 genes under positive selection in spermatogonia based on

enrichment of non-synonymous mutations in sperm[21]. Direct sequencing of sperm is effectively equivalent to assaying paternal de novo mutations across all genes in a large cohort of randomly selected trios barring those mutations not compatible with live birth, a case that is discussed below. We used these data to verify some of our predictions.

To evaluate the level of concordance, we tested whether CES driver candidates identified in human genetics datasets achieved significance in the test from ref. 21 aggregating all types of non-synonymous mutation[21]. Of all 40 genes identified here, 17 are nominally significant (20 at the *P* value threshold of 0.1 used in ref. 21) and 11 study-wide significant in ref. 21. We compiled the joint list of all putative CES genes between studies (Supplementary Table 20).

Next, we compared genes of LoF-1 and LoF-2 sets with 16 genes significant in the NanoSeq dataset exclusively due to LoFs (FDR 20%). As shown in Fig. 4a, out of 16 NanoSeq genes, 7 overlap with our LoF sets.

Partial overlap between the lists could be attributed to several factors. First of all, datasets obtained by NanoSeq have variable coverage across different genes, and indeed our CES candidates that are not significant in ref. 21 tend to have lower coverage (Extended Data Fig. 6). Second, for GoF-1 and LoF-1 the two approaches have differential statistical power. For the human genetics approach, the power is enhanced by the effect of CES drivers on NDD, whereas the power of the NanoSeq experiment is not boosted by the ascertainment. Finally, the power of both approaches is affected by sampling variance in the low mutation count data.

To demonstrate that the overlap is limited by power, we show that a residual CES effect exists in genes outside the overlap. We calculated the observed-to-expected ratio for LoF mutations in the NanoSeq dataset for genes that did not attain genome-wide significance, but are included in our LoF sets (Fig. 4b). We also measured the observed-to-expected ratio for genes outside the overlap using de novo variants in control trios (Fig. 4b). In both cases, we observe significant enrichment of LoF mutations ($P < 0.01$, Poisson test).

**Table 1 | Different data modalities offer possibilities for distinguishing types of modulator of observed de novo mutation rates**

| | NDD | Clonal expansions | | | Embryonic deleteriousness |
|---|---|---|---|---|---|
| | | Spermatogonia | Early development | Oocyte progenitors | |
| De novo effect in cases | ↑ | ↑ | ↑ | ↑ | ↓ |
| De novo effect in controls | 0 | ↑ | ↑ | ↑ | ↓ |
| Effect in sperm sequencing | 0 | ↑ | ↑ | 0 | 0 |
| Segregation in population | ↓ | ↑ | ↑ | ↑ | ↓ |
| **Other considerations** | | | | | |
| Disease phenotype | Yes | No | No | No | No |
| Somatic mosaicism | No | No | Yes | No | No |
| Parental bias | Baseline | Paternal | 50/50 | Maternal | Baseline |

In the NanoSeq dataset, the combined LoF-1 and LoF-2 gene sets demonstrate a 16.6-fold (Poisson 95% CI 13.5–20.2) excess of LoF mutations. For the GoF set, we observe a notable 524-fold excess of mutation counts in sperm sequencing (Poisson 95% CI 311–828).

## Discussion

The unique position of gametogenic tissues as a bridge between somatic and germline evolution allows CES to be viewed from two different perspectives and in terms of two different fields. From the biological perspective, CES can be related to cancer with similar genes being involved in clonal expansions in normal tissues and oncogenic transformation. From the human genetics perspective, CES acts as a factor inflating mutation rate at certain positions. Inflation of mutation rate leads to increase in population frequency of CES drivers. However, because the putative CES drivers reported here are kept at low frequency in the population, we expect that they are subject to strong negative selection even in cases with no obvious disease association.

### Biological function of CES drivers

Biological functions of many of our CES driver candidates are consistent with the role in clonal expansions (Supplementary Table 6). We note that the method used here biases the resulting gene sets towards NDD and ASD causal genes, but the functional roles of CES driver candidates are distinct from the bulk of known ASD–NDD genes.

Specifically, out of 40 identified genes, 20 have a role in major signalling pathways (Supplementary Tables 6 and 15–18). The pathways include MAPK, WNT and TGFβ (Supplementary Table 13). As expected, the MAPK pathway has the highest enrichment (eight genes, Fig. 4c). We noticed that the putative CES drivers are concentrated on the pathway structure and the LoF and GoF annotations are mirrored in the activator–repressor roles (Fig. 4c).

Serine-threonine kinase CSNK2A1 and genes activated by it (PACS1 and PACS2)[40] form an intriguing group of functionally related CES driver candidates within the GoF set. Another related kinase, CSNK2B (ref. 40), has been reported in ref. 21. The R203T mutation in PACS1 is the most recurrent variant in the NDD cohort[41] and explains 0.1% of all cases, and the E209K mutation in PACS2 explains another 0.04%. The high recurrence of these variants in the NDD cohort may be explained by their role in both NDD[41,42] and CES.

As expected for genes involved in clonal expansions, 12 out of 40 genes identified here are COSMIC census cancer drivers[28]. Four of them (*FGFR3*, *PTEN*, *MTOR* and *BRAF*) have been specifically associated with testicular tumours[43–46].

There is a discrepancy in the average elevations of mutation rate due to CES between GoF and LoF mutations (524-fold versus 17-fold). We propose two possible explanations. First, spermatogonia cells are diploid, and the variants discussed here are probably heterozygous. LoF mutations are often partially recessive with moderate effects in heterozygotes[47]. By contrast, GoF mutations are usually dominant. The second explanation comes from population genetics. We observe very limited numbers of GoF CES drivers in a single gene (Fig. 1e). On the other hand, because all LoF mutations within a gene have identical functional consequences, the number of potential CES driver mutations is large. As CES drivers are frequently involved in severe diseases, the GoF-like 500-fold elevation in mutation rate might generate a substantial rate of de novo pathogenic mutations, for example, if the same inflation was present for LoF mutations in *ARID1B*, about 0.5% of all individuals would be born with NDD if the CES effect was so high. Such pathogenic burden created by LoF mutations in a single gene will trigger efficient negative selection on the CES effect.

### Impact of CES drivers on NDD

Here, we develop a series of tests to identify CES drivers using the counts of de novo mutations in cohorts ascertained by disease, primarily NDD. CES effect and disease ascertainment independently increase the counts of de novo mutations in these cohorts. This gives us an upper hand to find CES genes with involvement in NDD compared with non-ascertained cohorts such as trio controls or sperm sequencing. Indeed, out of 40 recovered genes, 26 have orthogonal functional or genetic evidence of involvement in NDD. As expected based on the properties of the tests discussed above, GoF and LoF-1 gene sets are more enriched in genes causal for NDD. This falls in line with the notion of CES determining the prevalence of many types of NDD. A well-known example is given by Noonan syndrome in which clonal expansions underlie both aetiology and prevalence of the disease. For Noonan syndrome mutations, the effect observed in NanoSeq is 440-fold (Poisson 95% confidence interval 91–1285), which resolves the controversy between the net baseline mutation rate of roughly $10^{-7}$ and the prevalence of the syndrome of roughly $10^{-4}$. Prevalence of CES-related diseases such as Apert syndrome and achondroplasia scales exponentially with paternal age[8,48,49]. Therefore, it is expected that prevalence of any CES-related condition would similarly show the exponential dependency on the age of the father.

On the other hand, many of the recovered CES driver candidates, especially the ones from the LoF-2 set, for example, *MIB1* or *TCF7L2*, have no strong supporting evidence for NDD, meaning that they could be false positives in studies of gene-disease association that rely solely on de novo enrichment. However, these studies may find evidence of the causal role in NDD not confounded by CES such as: (1) transmission distortion and other types of familial segregation (Extended Data Fig. 7), (2) case–control analyses, (3) phenotypic similarity of mutation carriers and (4) functional assays in vitro and in vivo.

### Other modulators of mutation counts

Aside from CES and disease ascertainment, one modulator of the observed de novo counts is embryonic deleteriousness (partial lethality) (Table 1 and Supplementary Fig. 1). It is expected to decrease the

observed mutation rates and therefore renders our approaches for finding CES genes conservative. The comparison of de novo counts between sperm sequencing and trio sequencing has a potential to highlight lethal mutations. Following this logic, we find that LoF mutations in *RASA2* are consistent with embryonic deleteriousness.

Although here we interpret the elevation of mutation rate relative to the baseline as the effect of CES, the mutation rate increase may be due to clonal expansions in early development or in oocyte progenitors. Formally, our approach does not pinpoint the source of the clonal expansions. Still, substantial overlap (given statistical limitations) with sperm sequencing data supports the CES hypothesis. No mutations reported in ref. 21 have appreciable variant allele frequency in sperm[21], ruling out early developmental origin. Historically, CES is the only reported clonal mechanism responsible for the elevation of germline mutation rate.

In this study we have shown that the phenomenon of CES might be of unexpectedly great importance for future studies in the fields of genetics of rare disease, population genetics, cancer biology and in the emerging field of clonal evolution in somatic tissues.

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

## Methods

### Data

We used the dataset of de novo variants in probands affected by NDD published in ref. 9. For ASD, we used the de novo portion of the SFARI dataset published in ref. 12. For congenital heart diseases, encephalopathic epilepsy and trios with healthy offspring used as controls, we used aggregated datasets of de novo variants in cohorts with probands affected by the respective conditions[9–19,50–54], For any non-NDD and non-ASD cohorts, we filtered out probands with duplicate IDs and samples in which de novo variants were not obtained by whole genome or whole exome sequencing. Next, for the cohorts with de novo variants reported in hg19 coordinates, we mapped the coordinates to hg38 using LiftOver[55].

A recently published NanoSeq dataset of sperm cell sequencing was downloaded from the supplementary materials from ref. 21 and candidates due to LoF were taken as genes with the LoF enrichment at the FDR < 0.2 level given in the original table (qtrunc_cv column).

COSMIC cancer tiers of the putative CES genes and observed recurrence of variants in these genes with respect to cancer association, were obtained from COSMIC database on 25 October 2024 (ref. 28). The list of genes causing clonal expansions in haematopoietic tissues was obtained from ref. 3.

Data on single-cell gene expression in sperm progenitors and oocytes were downloaded from The Human Protein Atlas[56] and data on single-cell RNA sequencing of early human embryos was obtained from the supplementary information in ref. 57. The structure of the MAPK pathway shown on Fig. 4d was obtained from WikiPathways/WP382 (ref. 58). For the per-site mutation rate estimates and quality control tracks, we used Roulette-MR and Roulette-QUAL features of Roulette, respectively[25].

### Processing of genes

Because Roulette mutation rates are available only for autosomes, we restricted all analyses to autosomal genes. To avoid ambiguity in assigning variants, we further limited our dataset to genes with non-overlapping coding sequences. For any pair of genes with overlapping coding sequence regions, we retained the longer gene and excluded the shorter one. In cases in which more than two genes overlapped, we selected the largest subset of non-overlapping genes that maximized the total coding sequence length. This filtering yielded a final set of 17,791 autosomal protein-coding genes, listed in Supplementary Table 14.

### Processing of variants

We included only coding variants with a 'high' or 'TFBS' quality score in the Roulette-QUAL track. For each gene, variants were assigned to one of three functional categories. First, LoF variants were defined as those with both LOFTEE[20] high-confidence annotations and a variant effect predictor (VEP)[59] consequence of stop_gained, splice_donor_variant or splice_acceptor_variant. Second, synonymous variants were defined as those with a VEP annotation[59] containing only the term 'synonymous', excluding any variants (possibly) involved in splicing. Third, we included missense variants only if they were annotated as 'missense' in VEP and had AlphaMissense[60] scores greater than 0.1.

### Mutational expectation

To scale mutational expectations in an unbiased way, we normalized by the observed number of synonymous de novo mutations. This approach accounts for cohort-specific factors such as average parental age and sequencing coverage. For a given variant $v$, we define the expected number of occurrences in a de novo dataset as:

$$\lambda_v := N_S \frac{\mu_v}{\sum_{v \in S} \mu_v},$$

where $N_S$ is the total number of synonymous de novo mutations observed in the cohort, $S$ is the set of all admissible synonymous variants and $\mu_v$ is the unscaled Roulette mutation rate assigned to $v$. This formulation reflects the assumption that de novo mutation events occur independently, allowing expected counts to be proportional to the summed rates.

For any annotation $A = \{v\}$, the expected number of de novo mutations is:

$$\lambda_A := \sum_{v \in A} \lambda_v = N_S \frac{\sum_{v \in A} \mu_v}{\sum_{v \in S} \mu_v}. \tag{2}$$

This enables comparison of observed and expected mutation counts across functional categories while controlling for mutation rate variation and technical biases.

### Control for biases in Roulette

To test for overdispersion in Roulette estimates at the single-variant level, we compared two models of counts of de novo synonymous mutations in the NDD cohort: a Poisson model assuming no overdispersion, and a negative binomial model assuming overdispersion by a factor of $\gamma + 1$. Using expected mutation counts defined as in equation (2), the likelihood under the Poisson model is:

$$\mathcal{L}_0 = \prod_{v \in S} e^{-\lambda_v} \frac{\lambda_v^{n_v}}{n_v!}, \tag{3}$$

where $n_v$ is the observed number of recurrent de novo mutations at site $v$. The likelihood under the negative binomial model, with variance inflated by a factor of $\gamma + 1$, is:

$$\mathcal{L}_\gamma = \prod_{v \in S} \binom{n_v + \frac{\lambda_v}{\gamma} - 1}{n_v} \left( \frac{\gamma}{\gamma+1} \right)^{n_v} \left( \frac{1}{\gamma+1} \right)^{\frac{\lambda_v}{\gamma}}, \tag{4}$$

using the parametrization of the negative binomial distribution by mean $\lambda_v$ and overdispersion factor $\gamma$, such that the variance is $\lambda_v(1 + \gamma)$. As equation (4) can be rewritten as a Poisson–Gamma mixture, $\lambda_v/\gamma$ need not be an integer.

After maximizing the likelihood in equation (4) over $\gamma$, we obtain $\hat{\gamma} = 0.004$. Comparing the log-likelihoods of equations (3) and (4), we find a difference of $\log\mathcal{L}_\gamma - \log\mathcal{L}_0 = 31.1$. Applying the Akaike information criterion (AIC), which penalizes extra parameters by 2 log-likelihood units, this strongly supports the presence of modest but statistically significant overdispersion. As a result, we use the negative binomial distribution with $\gamma = 0.004$ rather than the Poisson model when modelling individual mutations in the GoF gene set.

Beyond single-site effects, Roulette estimates might have regionally correlated errors resulting in some regions having generally overpredicted or underpredicted mutation rates. In particular, correlated errors in residuals could systematically distort the expected counts for specific regions of interest, such as protein-coding genes and may be undetectable at the per-site level. To assess this, we tested for overdispersion in the number of synonymous de novo mutations aggregated by gene. Let $n_g$ be the observed and $\lambda_g$ the expected count of synonymous mutations in gene $g$, with $\lambda_g$ obtained by summing site-specific rates as in equation (2). We then substitute $n_v \rightarrow n_g$ and $\lambda_v \rightarrow \lambda_g$ in equations (3) and (4), and re-estimate the log-likelihoods.

In this gene-level analysis, the log-likelihood difference is $\log\mathcal{L}_\gamma - \log\mathcal{L}_0 = 0.69$, which falls well below the AIC threshold for including an extra parameter. Thus, we find no evidence of significant variance inflation at the gene level.

To further test for overdispersion in the gene-level expectations $\lambda_g$, we used a regression-based approach. Applying the law of total variance:

$$\mathrm{Var}(n_g) = \mathbb{E}[\mathrm{Var}(n_g|\lambda_g)] + \mathrm{Var}[\mathbb{E}(n_g|\lambda_g)],$$

where the first term captures residual (Poisson) noise and the second term reflects underlying variation in $\lambda_g$. As $\mathbb{E}(n_g|\lambda_g) = \lambda_g$ and $\mathrm{Var}(n_g|\lambda_g) = \lambda_g$ under the Poisson model, we expect:

$$\mathrm{Var}(n_g) = \mathbb{E}(\lambda_g) + \mathrm{Var}(\lambda_g).$$

To test this, we computed sample estimates of each term and tested the Poisson assumption with the chi-squared goodness-of-fit test on statistic

$$\chi^2 = \sum_g \frac{(n_g - \lambda_g)^2}{\lambda_g},$$

with degrees of freedom equal to the number of genes minus 1 (17,790). The resulting $P$ value of 0.6 confirms consistency with the Poisson model, with no evidence of excess dispersion. We thus conclude that gene-level Roulette residuals are effectively uncorrelated, and that variance inflation in gene-level mutational expectations is negligible.

Consequently, for gene-level aggregation of variant counts in LoF-1 and LoF-2 sets, we used Poisson tests to assess enrichments.

## Statistics

**Phenotypic homogeneity in the NDD cohort.** Our procedures yielding LoF-1 and GoF sets of putative CES drivers assume uniform phenotypic sampling across the NDD cohort (Main text and below). Deviations from uniformity—for example, overrepresentation of certain syndromes—could result in differential ascertainment of associated mutations, potentially inflating effect sizes for certain genes and introducing false positives to LoF-1 and LoF-2 sets.

To assess homogeneity of recruitment in the composite NDD cohort used here, we compared contributions of the three subcohorts (DDD, GeneDx and RUMC) comprising it to de novo LoF variation in individual genes. If the cohorts were sampled in a uniform way, the fractions of de novo variants coming from each cohort across all genes should correspond to the per-cohort mutation rate, which is proportional to the number of synonymous mutations per individual cohort (equation (2)). The numbers of variants in each gene should therefore come from the multinomial distribution with probabilities given by per-cohort synonymous variant counts and the number of events given by the number of observed de novo variants.

We tested this hypothesis by comparing four models: (1) a multinomial model with probabilities fixed by observed synonymous variant counts (zero free parameters); (2) a multinomial model with unconstrained probabilities to test for shifts in representation (two free parameters due to three probabilities that need to sum up to 1); (3) a Dirichlet-multinomial model centred on the observed synonymous proportions, allowing for overdispersion (one free parameter: amplitude of Dirichlet) and (4) a fully flexible Dirichlet-multinomial model accounting for both overdispersion and shifted means (three free parameters: two for unconstrained multinomial and one more parameter for the amplitude of Dirichlet).

For each model, we computed the total likelihood across all genes and compared models using the AIC. The unbiased multinomial model (model (1)) was the one preferred by AIC, indicating that the observed gene-level LoF variant distributions are consistent with uniform cohort sampling.

To further validate that our LoF-1 and LoF-2 sets are not confounded by non-uniform sampling, we repeated the analysis on specifically LoF-1 and LoF-2 putative CES drivers. In both cases, the AIC again favoured the null multinomial model. Together, these results suggest that recruitment across the DDD, GeneDx and RUMC cohorts was phenotypically homogeneous with respect to LoF variant ascertainment. Full likelihood

and AIC values for all model comparisons are reported in Supplementary Table 2.

**Prevalence of NDD.** The identification of putative CES drivers in the LoF-1 and GoF sets relies on the assumption that the disease prevalence estimate $P(D)$ used in equation (1) is not inflated. Overestimating $P(D)$ would lead to underestimating the maximum plausible effect size attributable to ascertainment, thereby increasing the risk of false-positive CES calls in highly penetrant genes.

Epidemiological estimates of NDD prevalence—based on diagnoses of intellectual disability or global developmental delay—typically fall within the range of 2–3% (ref. 26). To ensure a conservative interpretation, we imposed a lower bound of 1% on $P(D)$, which corresponds to an upper bound of 100 on the effect of ascertainment.

One further consideration suggests that 1% is an appropriate lower boundary for NDD prevalence. Almost all significantly mutated genes have the effects below 100 with very few genes greatly exceeding this threshold (Fig. 3d). The possibility that the prevalence is even lower than 1% and the observed distribution of gene effects is fully explained by disease ascertainment is inconsistent with the existence of genes with substantial penetrance for NDD.

**Construction of the GoF set.** To define the GoF set of putative CES driver genes, we applied a negative binomial test. The expected count of de novo variants at position $v$ was set to $\mu_v/P^*(D)$, where $\mu_v$ is the Roulette mutation rate at $v$ and $P^*(D)$ is the lower bound on disease prevalence, fixed at 0.01 (Main text). To account for overdispersion at the single-site level, we used a variance inflation parameter $\gamma = 0.004$, as defined in equation (4). For multiple testing correction, we applied a Bonferroni adjustment using a correction factor equal to the total number of autosomal missense variants annotated by VEP and filtered to have an AlphaMissense score above 0.1 (49,686,008 tests).

**Construction of the LoF-1 set.** To define the LoF-1 set of putative CES driver genes, we applied a Poisson test to LoF de novo variants observed in the NDD cohort. The expected count of variants in gene $g$ was set to $\lambda_g/P^*(D)$, where $\lambda_g$ is the LoF mutation rate for gene $g$ as defined in equation (2) and $P^*(D)$ is the lower bound on disease prevalence, fixed at 0.01 (Main text). As in the GoF analysis, this expectation corresponds to the number of events expected under maximal disease ascertainment (that is, full penetrance). The choice of the Poisson model is justified in the 'Control for biases in Roulette' section above. For multiple testing correction, we used a Bonferroni factor equal to the number of autosomal genes with at least one filtered LoF site (17,791 tests). This yielded 5 Bonferroni-significant genes at the 0.05 level, and 1 more gene (*FOXG1*) significant at FDR < 0.1, using the Benjamini–Hochberg procedure (Fig. 2a,d).

**Construction of the LoF-2 set.** The LoF-2 set of putative CES driver genes was constructed using the combined ASD + NDD cohort in two steps. First, we identified genes with a significant excess of de novo LoF variants relative to expectation ($n_g > \lambda_g$), where $n_g$ is the observed LoF count and $\lambda_g$ is the Roulette-based expectation for gene $g$. As in the LoF-1 analysis, we used a Poisson test with correction for 17,791 tested autosomal genes (Supplementary Table 14). On the second step, we defined the LoF-2 set as those genes passing a FDR threshold of less than 0.1 and having gnomAD-v4 LOEUF scores greater than 0.5 (Fig. 2a).

**Excess of LoF variants in NDD compared with NanoSeq.** To test whether the excess of de novo LoF variants observed in the NDD cohort exceeded that in the sperm NanoSeq dataset, we used a Binomial test. For each gene $g$, we set the probability of success to the expected fraction of LoF mutations arising from NanoSeq, defined as $\frac{\lambda_{m,g}}{\lambda_{m,g} + \lambda_n}$, where $\lambda_{m,g}$ is the expected LoF count in NanoSeq and $\lambda_n$ is the expected LoF count in NDD. The number of trials was set to $n_g + m_g$, where $n_g$ and $m_g$

are the observed LoF counts in gene $g$ in the NDD and NanoSeq datasets, respectively. The test compared the observed fraction $\frac{m_g}{m_g + n_g}$ with the expected null fraction. Only genes with at least one LoF variant in both datasets were tested. A gene was deemed significant if its $P$ value, Bonferroni-corrected for the number of tests ($N = 5$), was below 0.05. This test identified only *CTNNB1* and *PTEN* as significantly enriched for LoF variants in the NDD cohort relative to NanoSeq. Because the NanoSeq donors were older on average than the parents of individuals in the NDD cohort, this comparison is conservative.

The test for the excess of variants in NanoSeq compared with the excess in the NDD cohort highlights *CUL3* and *ARID1A* as genes with a larger excess in NanoSeq. Although this effect may be confounded by the age differences between the parents in the NDD cohort and the donors of samples used in the NanoSeq experiment, an alternative explanation may be embryonic deleteriousness of LoF variants in these genes.

## Expression and gene set enrichment analyses

Expression analysis of candidate genes was performed using single-cell data from The Human Protein Atlas. For each gene, the Atlas reports transcripts per million (TPM) as a measure of expression across a wide range of tissues. To account for gene-specific differences in baseline expression, we normalized the TPM values in spermatogonia and oocytes by the maximum TPM observed for each gene across all tissues. To statistically assess differences in expression patterns between CES driver candidates and other genes, we applied the Mann–Whitney $U$-test to the normalized TPM values. Gene set enrichment analysis was performed using the g:Profiler[27] web service with default parameters. We tested five gene sets: the GoF, LoF-1 and LoF-2 sets individually; the union of LoF findings (LoF-1 and LoF-2); and the union of all three sets. Full g:Profiler results are provided in Supplementary Tables 15–18.

## GeneBayes update and CES

To obtain estimates of LoF constraint in the LoF-2 set of genes unimpacted by mutation rate misspecification we used prior estimates of the selection coefficient against heterozygotes, produced by the GeneBayes method. These estimates result from a prior distribution fit to all LoF variation using a set of global gene features and should be minimally impacted by variation in any one gene[39]. To assess the robustness of the resulting list of genes to the constraint estimation, we also performed an orthogonal analysis of differences between prior and posterior GeneBayes $s_{\mathrm{het}}$ estimates, which were obtained on gnomAD-v3. As expected under moderate selection against LoF mutations and CES, prior estimates of $s_{\mathrm{het}}$ were substantially higher than those estimated using polymorphism alone. The rationale here is that, given downwardly biased mutation rate estimates of CES genes, the data should shift the prior down by factors larger than those expected for other genes. This analysis recapitulates our results obtained with LOEUF.

Owing to the demographic complexity of the gnomAD-v4 sample, we used a data-driven approach to estimate $s_{\mathrm{het}}$ for each gene. We first fit how both the proportion of polymorphic sites as well as the shape of the site frequency spectrum depend on $s_{\mathrm{het}}$, conditional on $\mu$. For this, we fit a multinomial model to the observed allele counts using prior estimates of as a covariate. To ensure that the site frequency spectrum (SFS) was not too sparse at higher allele counts, we binned sites by allele count with boundaries [0,1,2,3,4,5,6,7,8,9,10,11,16,24,36,100]. The observed distribution of alleles counts in each Roulette mutation rate bin was then transformed to multinomial coefficients such that

$$\log \frac{P(K = i \mid \mu)}{P(K = 0 \mid \mu)} = \beta_i^{\mu} \tag{5}$$

This model for the neutral SFS in each bin was first estimated using synonymous SNV counts. Next, we fit a model to all LoF polymorphism such that

$$\log \frac{P(K = i \mid \mu)}{P(K = 0 \mid \mu)} = \beta_i^{\mu} - \beta_i^{s} \sqrt{s_{\mathrm{het}}}. \tag{6}$$

The $\beta_i^{\mu}$ coefficients were determined by the synonymous SFS and $\beta_i^{s}$ coefficients were fit with maximum likelihood using the L-BFGS-B algorithm as implemented in scipy. The square root dependency on was determined by manual inspection of the SFS, binned by $s_{\mathrm{het}}$ and $\mu$. The values used to fit this global multinomial model were prior mean values output by GeneBayes. This procedure yielded a model for the SFS in each mutation rate bin, given a value of $s_{\mathrm{het}}$. We then re-estimated for LoF mutations in each gene using this model using maximum likelihood and bounding to [0,1].

## Segregation of LoF-2 polymorphism in the human population

LoF-2 genes, despite showing a significant excess of LoF mutations in the NDD cohort, harbour many LoF variants segregating in the general population. This is reflected, for instance, in their relatively high LOEUF values. The number of segregating sites in a gene is governed by both mutation rate and selection. If these genes are genuine CES drivers, we expect an elevated mutation rate to contribute to the number and frequency of LoF SNVs. In such cases, both the number of segregating sites and their allele frequencies would appear higher than expected under a fixed selection coefficient.

In a cohort with more than 1 million chromosomes, this hypothesis can be evaluated by comparing the empirical expectation and variance of LoF allele frequency distributions across genes. Whereas both mutation rate and selection influence the mean and variance of allele frequencies, recurrent mutation produces alleles that tend to be more common but show lower variance in frequency, due to reduced genetic drift. By contrast, relaxed selection increases both mean and variance via enhanced drift. Thus, we expect genes with elevated mutation rate to show lower variance in allele frequency relative to their mean than genes under relaxed selection.

To formalize this, we applied the gamma-Poisson approximation for allele frequency spectra under strong selection and recurrent mutation[38].

**Gene filtering.** For this analysis, we retained only genes with at least 10 LoF sites passing Roulette-QUAL quality control and having unscaled Roulette mutation rate $\mu > 1$ (for example, CpG>TpG transitions). This filtering ensures reliable estimates of the mean and variance of allele frequencies for each gene.

**Model.** We assume a three-step generative model for LoF allele counts in each gene:
(1) Mutations occur randomly at any LoF site in the gene.
(2) Each derived variant segregates in a diploid population of effective size $N_{\mathrm{e}}$, experiencing negative selection of intensity $s$ in heterozygotes. We assume strong selection and no lineage interference.
(3) A sample of $n$ alleles is drawn from the population.

Let $k_{\upsilon}$ denote the number of observed derived LoF variants $\upsilon$ in the population sample. In what follows, we will drop the index $\upsilon$. We assume the variance of $k$ to be the sum of contributions of the three assumed steps without any extra terms:

$$\mathrm{Var}(k) = \mathrm{Var}_{\mathrm{sampling}}(k) + \mathrm{Var}_{\mathrm{seg}}(k) + \mathrm{Var}_{\mu}(k), \tag{7}$$

where the components correspond to mutation heterogeneity across sites, drift-induced segregation, and Poisson sampling noise, respectively. Under the Gamma approximation introduced in ref. 38, the distribution of allele frequencies at LoF sites is modelled with $4N_{\mathrm{e}}\mu$ as the shape parameter and $1/(4N_{\mathrm{e}}s)$ as the scale parameter. The expectation of $k$ is then:

$$\mathbb{E}(k) = \frac{n\mathbb{E}(\mu)}{s}.$$

And the first two terms of equation (7) are given by the variance of the Poisson and Gamma components, respectively:

$$\mathrm{Var}_{\mathrm{sampling}}(k) \equiv \mathbb{E}(k) = \frac{n\mathbb{E}(\mu)}{s},$$

$$\mathrm{Var}_{\mathrm{seg}}(k) = \frac{n\mathbb{E}(\mu)}{s}\frac{n}{4N_e s}.$$

The final term of equation (7) may be obtained by noticing that $\mathrm{Var}_{\mathrm{sampling}}(k) + \mathrm{Var}_{\mathrm{seg}}(k) = \mathbb{E}_\mu[\mathrm{Var}(k|\mu)]$. Applying the law of total variance,

$$\mathrm{Var}(k) = [\mathrm{Var}_{\mathrm{sampling}}(k) + \mathrm{Var}_{\mathrm{seg}}(k)] + \mathrm{Var}_\mu(k)$$
$$= \mathbb{E}_\mu[\mathrm{Var}(k|\mu)] + \mathrm{Var}_\mu[\mathbb{E}(k|\mu)]$$
$$= \frac{n\mathbb{E}(\mu)}{s}\left(1 + \frac{n}{4N_e s}\right) + \left(\frac{n}{s}\right)^2 \mathrm{Var}(\mu)$$
$$= \frac{n\mathbb{E}(\mu)}{s} + \frac{n\mathbb{E}(\mu)}{s}\frac{n}{4N_e s} + \left(\frac{n}{s}\right)^2 \mathrm{Var}(\mu),$$

$$\mathrm{Var}_\mu(k) = \left(\frac{n}{s}\right)^2 \mathrm{Var}(\mu).$$

We now model the effect of CES as a multiplicative factor $\kappa \geq 1$ on the mutation rate so that:

$$\mathbb{E}(k) = \kappa \times \frac{n\mathbb{E}(\mu)}{s}, \tag{8}$$

$$\mathrm{Var}(k) = \kappa \times \frac{n\mathbb{E}(\mu)}{s}\left(1 + \frac{n}{4N_e s}\right) + \kappa^2 \times \left(\frac{n}{s}\right)^2 \mathrm{Var}(\mu). \tag{9}$$

The final term in equation (9) containing dependency on $\kappa^2$ can be estimated from the regression of $k$ on $\mu$ with Poisson error. Subtracting this term from $\mathrm{Var}(k)$ leaves only the sample estimate of $\mathbb{E}_\mu(\mathrm{Var}(k|\mu))$.

## Estimating selection

This framework yields two estimators of $1/s$:
(1) From the mean allele count (equation (8)):

$$\frac{\kappa}{\hat{s}} = \frac{\bar{k}}{n\bar{\mu}}, \tag{10}$$

where $\bar{\mu}$ and $\bar{k}$ are the mean mutation rate and mean LoF allele count per gene.
(2) From the conditional variance (equation (9)), subtracting sampling noise:

$$\frac{\mathrm{Var}_{\mathrm{seg}}(k)}{\mathrm{Var}_{\mathrm{sampling}}(k)} = \frac{\mathbb{E}_\mu[\mathrm{Var}(k|\mu)] - \mathrm{Var}_{\mathrm{sampling}}(k)}{\mathrm{Var}_{\mathrm{sampling}}(k)} = \frac{n}{4N_e s}. \tag{11}$$

Note that only the first estimator (equation (10)) depends on $\kappa$ and is therefore sensitive to CES effects.

Figure 3e plots the log estimate of $\frac{n}{4N_e s}$ (excess variance due to drift) against the log estimate of $\frac{\kappa}{s}$ (elevated mean allele count due to CES). Because $\log(n/4N_e) - \log(s)$ is a constant offset from $\log(\kappa) - \log(s)$, genes with elevated mutation due to CES appear as positive residuals from the regression line.

This approach is conservative: the gamma-Poisson approximation is invalid for genes under weak or no selection, where allele frequency variance is even higher. Thus, such genes would not mimic the CES signal.

True NDD genes deviate negatively from the regression line, having lower mean allele frequencies than predicted under a fixed $s$. This is consistent with either sampling bias in gnomAD-v4 resulting from difficulties in recruitment of individuals affected with NDD or from selection acting against LoF variants in these genes even prenatally, leading to inherently reduced representation in population datasets.

*MIB1* **example.** For *MIB1*, we used Poisson regression to quantify the dependence of LoF allele frequency on mutation rate. To visualize this relationship (Fig. 3d), we binned sites by scaled mutation rate, merging bins with fewer than ten sites with the next-highest bin.

## Reporting summary

Further information on research design is available in the Nature Portfolio Reporting Summary linked to this article.

## Data availability

All data used in this study, including intermediate processed datasets, are provided in Supplementary Tables 1–20. Associated summary files are also available on Zenodo (https://zenodo.org/records/15660433)[61].

## Code availability

The full analysis pipeline, including code to reproduce figures and statistical analyses, is available on GitHub at https://github.com/mikemoldovan/CES_Discovery.

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

**Acknowledgements** We thank C. Boix, C. Chiang and R. Stana as well as A. Quinlan and J. Kunisaki for valuable discussions and helpful suggestions that improved the analyses presented in this study. This work was supported by the National Institutes of Health through grant nos. R35GM12713, R01MH101244 and U01HG012009.

**Author contributions** V.S., M.A.M. and S.S. jointly conceived the study and developed the methodological framework. V.S., M.A.M., E.K., P.K. and M.D.C.N. performed data analysis and interpreted results. V.S., M.A.M., E.K., P.K., M.D.C.N., R.R. and S.S. collaboratively drafted and revised the paper. S.S., R.R., V.S. and M.A.M. jointly supervised the project. All authors reviewed and approved the final version of the paper.

**Competing interests** The authors declare no competing interests.

**Additional information**
**Correspondence and requests for materials** should be addressed to Vladimir Seplyarskiy, Mikhail A. Moldovan or Shamil Sunyaev.

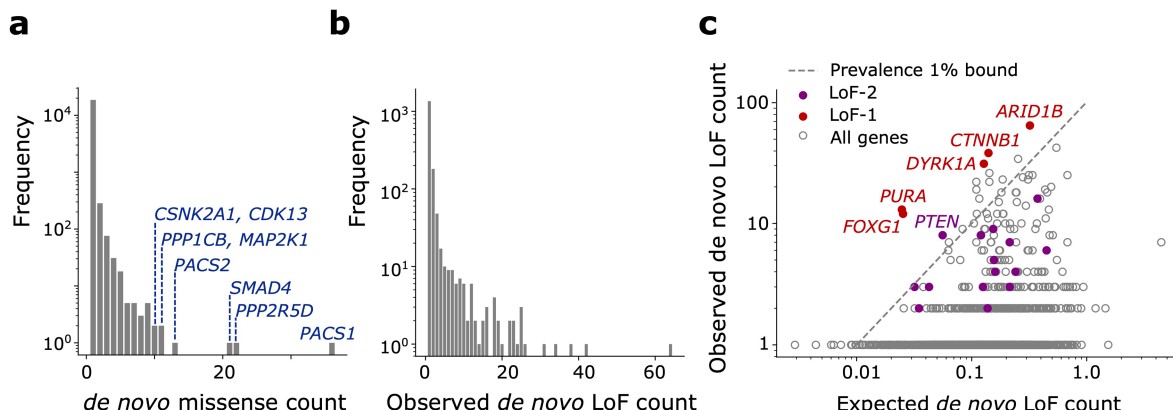

**Extended Data Fig. 1 | Counts of de novo variants in the NDD cohort.**
**(a)** Numbers of de novo missense variants stratified by recurrence. Genes harboring variants occurring >10 times in the cohort are shown in blue. **(b)** Numbers of de novo loss-of-function variants aggregated by gene stratified by recurrence. **(c)** Scatter plot of observed vs. expected de novo loss-of-function variant counts in the NDD cohort. LoF-1 set genes are shown in red; LoF-2 set genes are shown in purple. Upper bound of disease ascertainment in Eq. (1) given by the lower bound of prevalence of 1% is shown as a dashed line.

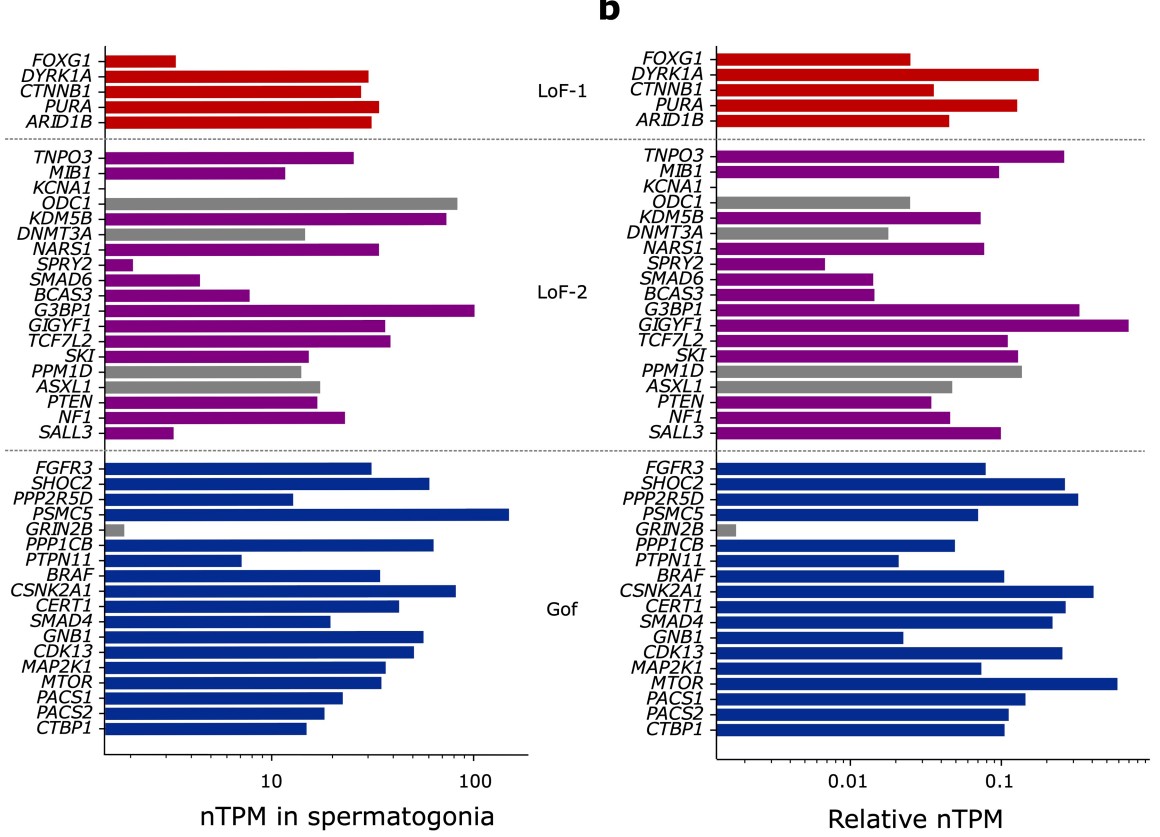

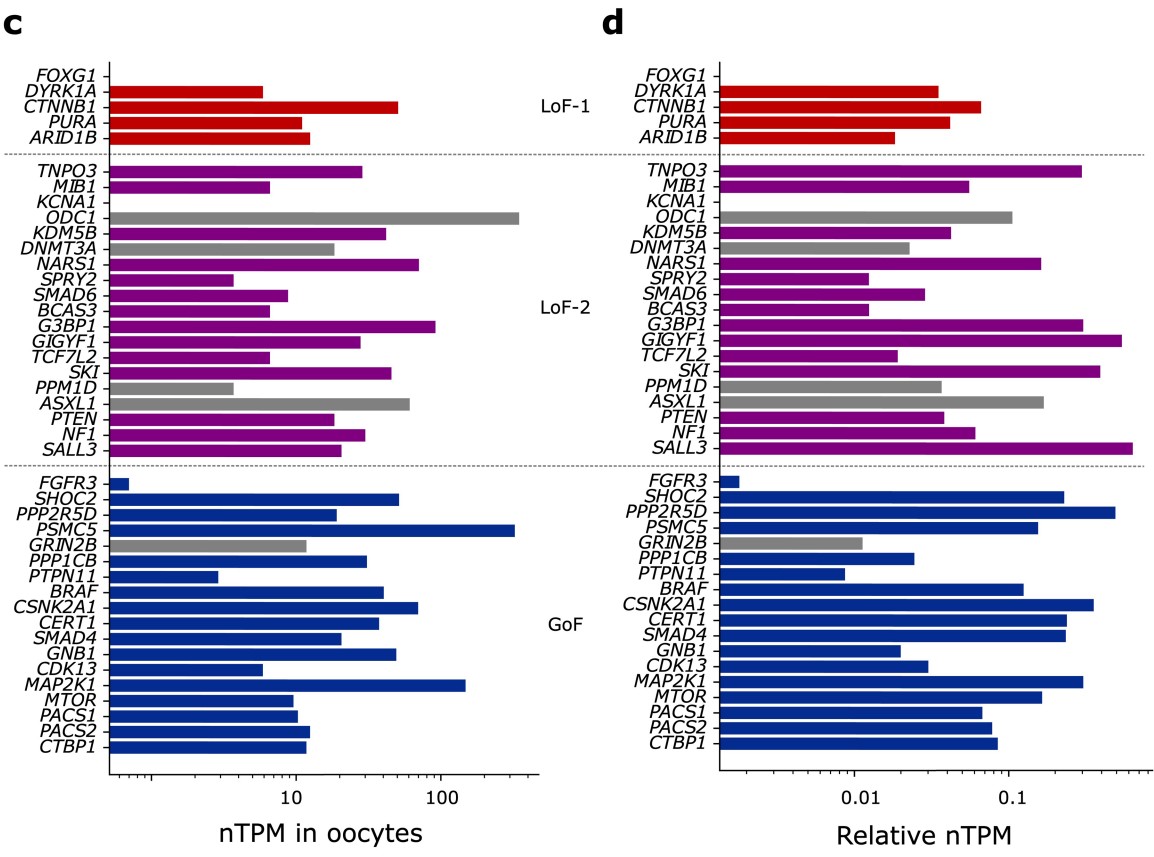

**Extended Data Fig. 2 | Expression of the identified CES drivers in germline tissues. (a)** nTPM values for spermatogonia reported in The Human Protein Atlas single-cell dataset. **(b)** nTPM values normalized by the maximal expression across all tissues for each gene. **(c)** nTPM values in oocytes. **(d)** Normalized nTPM values in oocytes.

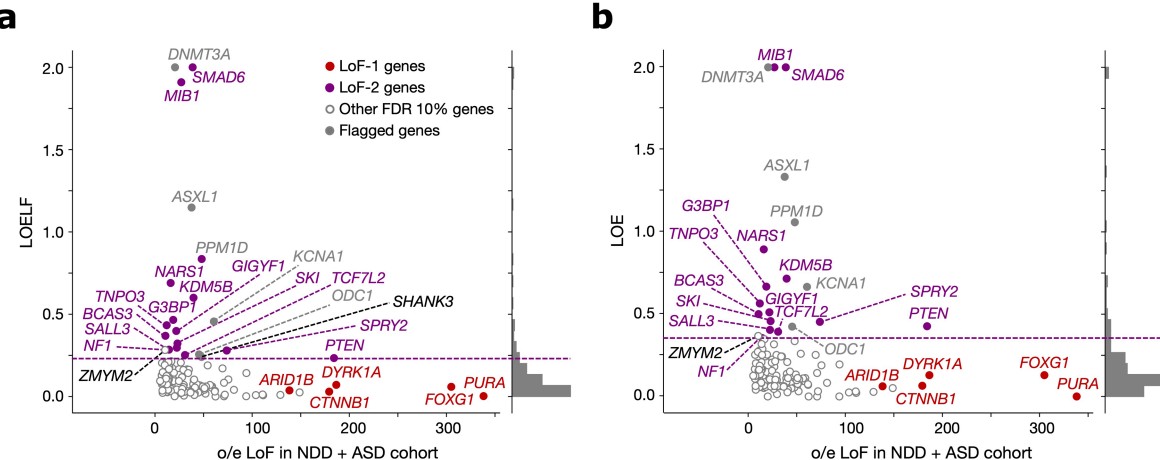

**Extended Data Fig. 3 | Stability of LoF-2 set with respect to the metric of loss-of-function constraint. (a)** Observed-to-expected variant count ratio (o/e) for de novo LoFs in genes with FDR < 0.1 in the neurodevelopmental disorder cohort (NDD) merged with the autism spectrum disorder (ASD) cohort plotted against the Loss-of-function Observed/Expected Lower-bound Fraction (LOELF) scores. The dashed violet line indicates the minimal LOELF value across LoF-2 genes of 0.23. LoF-2 genes are shown in violet, LoF-1 genes are shown in red, genes above the chosen LOELF threshold but not included in the LoF-2 set (*SHANK3* and *ZMYM2*) are shown in black. **(b)** Same as in (a), but for the Loss-of-function Observed/Expected (LOE) metric. The upper bound for LOE (shown as violet dashed line) is 0.355.

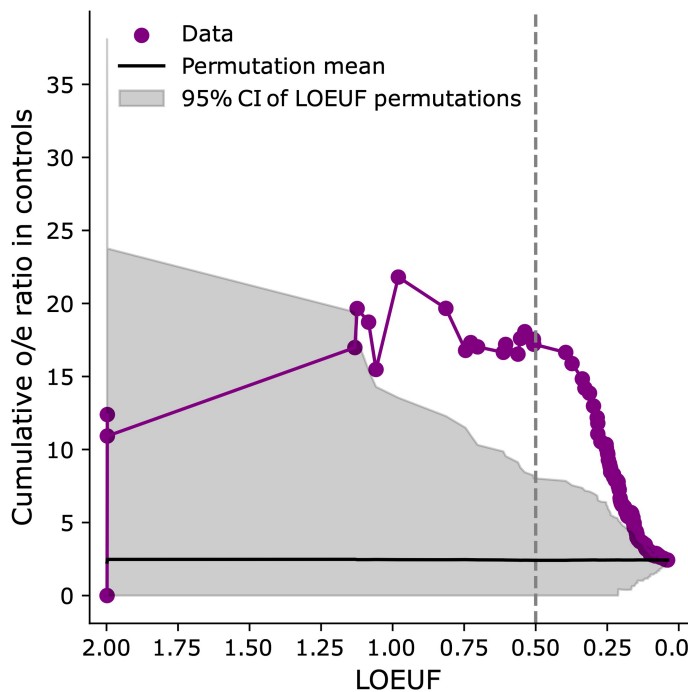

**Extended Data Fig. 4 | Ratio of observed-to-expected counts of LoF de novo mutations in a cohort of control trios for LoF-2 set genes.** The ratio is shown as a function of the LOEUF threshold: we aggregate all genes with LOEUF values lower than the value indicated on the x-axis and calculate the cumulative observed-to-expected ratio. The shaded grey area represents the 95% confidence interval obtained by permuting the LOEUF labels.

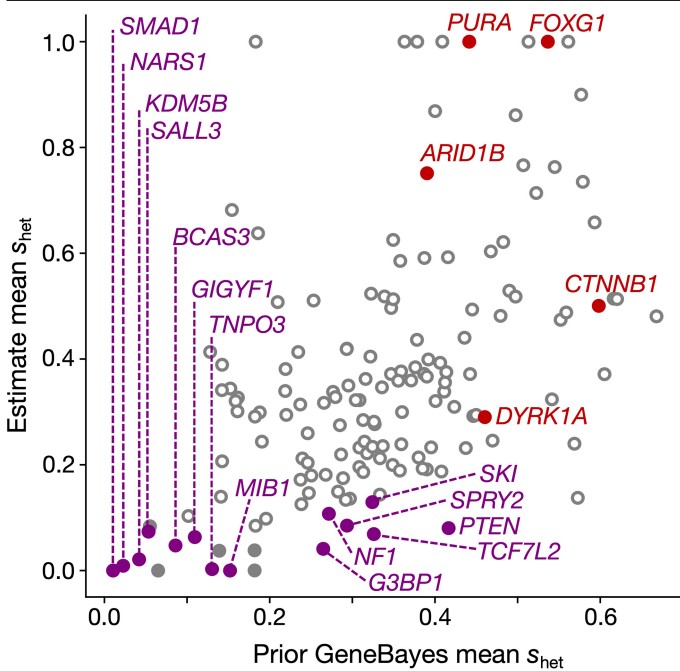

**Extended Data Fig. 5 | Validation of LoF-2 genes with a non-LOE metric.** Prior of the GeneBayes $s_{het}$ calculated using biological features of genes (x-axis) and the $s_{het}$ values updated with LoF polymorphism data from gnomAD-v4 (y-axis). See section 'GeneBayes update and CES' for details.

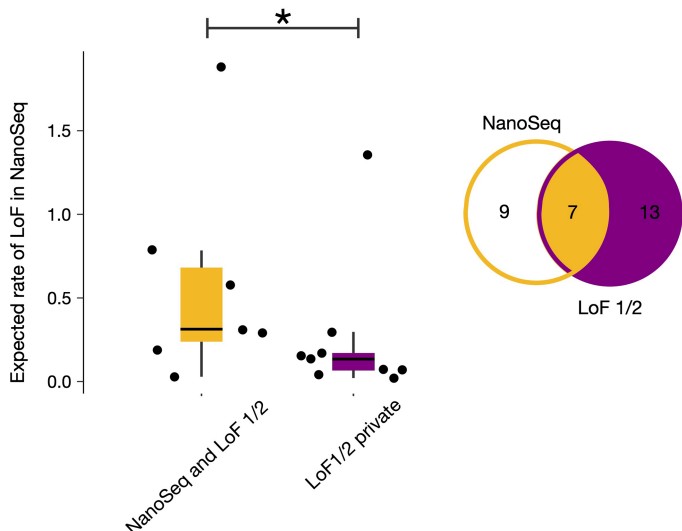

**Extended Data Fig. 6 | Expected LoF rate in NanoSeq data for LoF-1 and LoF-2 genes.** Rates are shown separately for genes overlapping with those significant in NanoSeq and for private LoF-1/2 genes. An asterisk (*) indicates $p < 0.05$ from the Mann–Whitney U test.

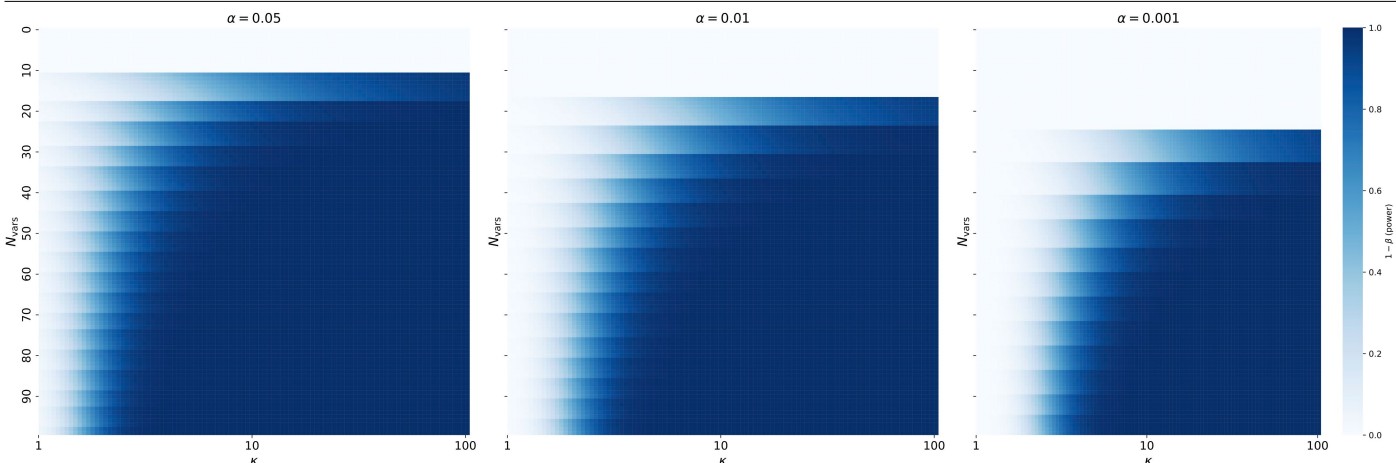

**Extended Data Fig. 7 | Power analysis of paternal transmission.** Statistical power (i.e., the probability of correctly detecting a signal when it exists calculated as the complement of type-2 error rate ß) of the Binomial test for paternal overtransmission relative to the baseline of 0.75 is shown across the range of CES-related mutation rate inflations κ and counts of observed variants. Results are presented for three significance levels: 0.05, 0.01, and 0.001.

# Reporting Summary

## Statistics

For all statistical analyses, confirm that the following items are present in the figure legend, table legend, main text, or Methods section.

| n/a | Confirmed | |
|---|---|---|
| ☐ | ☒ | The exact sample size (*n*) for each experimental group/condition, given as a discrete number and unit of measurement |
| ☒ | ☐ | A statement on whether measurements were taken from distinct samples or whether the same sample was measured repeatedly |
| ☐ | ☒ | The statistical test(s) used AND whether they are one- or two-sided<br>*Only common tests should be described solely by name; describe more complex techniques in the Methods section.* |
| ☒ | ☐ | A description of all covariates tested |
| ☐ | ☒ | A description of any assumptions or corrections, such as tests of normality and adjustment for multiple comparisons |
| ☐ | ☒ | A full description of the statistical parameters including central tendency (e.g. means) or other basic estimates (e.g. regression coefficient) AND variation (e.g. standard deviation) or associated estimates of uncertainty (e.g. confidence intervals) |
| ☐ | ☒ | For null hypothesis testing, the test statistic (e.g. *F*, *t*, *r*) with confidence intervals, effect sizes, degrees of freedom and *P* value noted<br>*Give P values as exact values whenever suitable.* |
| ☒ | ☐ | For Bayesian analysis, information on the choice of priors and Markov chain Monte Carlo settings |
| ☒ | ☐ | For hierarchical and complex designs, identification of the appropriate level for tests and full reporting of outcomes |
| ☐ | ☒ | Estimates of effect sizes (e.g. Cohen's *d*, Pearson's *r*), indicating how they were calculated |

*Our web collection on statistics for biologists contains articles on many of the points above.*

## Software and code

Policy information about availability of computer code

| Data collection | No software for data collection has been used |
|---|---|
| Data analysis | For data analysis, we used custom R and Python code that is deposited online under the MIT license at: https://github.com/mikemoldovan/CES_Discovery |

For manuscripts utilizing custom algorithms or software that are central to the research but not yet described in published literature, software must be made available to editors and reviewers. We strongly encourage code deposition in a community repository (e.g. GitHub). See the Nature Portfolio guidelines for submitting code & software for further information.

## Data

Policy information about availability of data

All manuscripts must include a data availability statement. This statement should provide the following information, where applicable:

- Accession codes, unique identifiers, or web links for publicly available datasets
- A description of any restrictions on data availability
- For clinical datasets or third party data, please ensure that the statement adheres to our policy

We used the following publicly available datasets:
Genetic variation in the human population reported in gnomAD-v4: https://gnomad.broadinstitute.org/news/2023-11-gnomad-v4-0/

Datasets of de novo variants in cohorts of patients affected with a variety conditions published as supplementary information in Refs 9,12,13,15,19,50-54
Sperm sequencing dataset published in the preprint by Neville et al., 2024 (DOI: 10.1101/2024.10.30.24316414v1)

## Research involving human participants, their data, or biological material

Policy information about studies with human participants or human data. See also policy information about sex, gender (identity/presentation), and sexual orientation and race, ethnicity and racism.

| | |
|---|---|
| Reporting on sex and gender | N/A |
| Reporting on race, ethnicity, or other socially relevant groupings | N/A |
| Population characteristics | N/A |
| Recruitment | N/A |
| Ethics oversight | N/A |

Note that full information on the approval of the study protocol must also be provided in the manuscript.

# Field-specific reporting

Please select the one below that is the best fit for your research. If you are not sure, read the appropriate sections before making your selection.

☒ Life sciences ☐ Behavioural & social sciences ☐ Ecological, evolutionary & environmental sciences

For a reference copy of the document with all sections, see nature.com/documents/nr-reporting-summary-flat.pdf

# Life sciences study design

All studies must disclose on these points even when the disclosure is negative.

| | |
|---|---|
| Sample size | Numbers of samples in the used datasets are as reported in the original studies |
| Data exclusions | We applied Roulette variant-quality filters (described in Ref 24) to censor positions with low quality of mutation rate predictions. Otherwise, no data were excluded. |
| Replication | As our study is theoretical in nature, experimental replication is not applicable here. To ensure computational reproducibility, we have created a GitHub repository containing all the relevant code (as stated in the "data availability statement" and "code availability statement") |
| Randomization | As no experiments were designed, no randomization procedures were implemented. Nor was there any randomized subsampling of the repurposed data and no random bootstrap- permutation-based statistical tests which could be considered a form of randomization. |
| Blinding | As no human decision-based procedures were employed, blinding is not applicable in this case. |

# Reporting for specific materials, systems and methods

We require information from authors about some types of materials, experimental systems and methods used in many studies. Here, indicate whether each material, system or method listed is relevant to your study. If you are not sure if a list item applies to your research, read the appropriate section before selecting a response.

### Materials & experimental systems

| n/a | Involved in the study |
|---|---|
| ☒ | ☐ Antibodies |
| ☒ | ☐ Eukaryotic cell lines |
| ☒ | ☐ Palaeontology and archaeology |
| ☒ | ☐ Animals and other organisms |
| ☒ | ☐ Clinical data |
| ☒ | ☐ Dual use research of concern |
| ☒ | ☐ Plants |

### Methods

| n/a | Involved in the study |
|---|---|
| ☒ | ☐ ChIP-seq |
| ☒ | ☐ Flow cytometry |
| ☒ | ☐ MRI-based neuroimaging |

# Plants

Seed stocks

N/A

Novel plant genotypes

N/A

Authentication

N/A

