## [Peer Review File · Nature]

Hotspots of human mutation point to clonal expansions in spermatogonia

Corresponding Author: Professor Shamil Sunyaev

Version 0:

Reviewer comments:

Referee #1

(Remarks to the Author)

In this manuscript, the authors use a diverse set of approaches to identify genes of functional importance with more loss-of-function mutations (or missense mutations) than expected from an advanced prediction of site-specific mutation rates to identify genes that have the capacity to cause clonal expansion in spermatogonia of the male germline (and thus appear to have a higher mutation rate). We already know from a few well-studied cases that such clonal expansion can be very detrimental since the mutations causing them are more likely to cause syndromes in the next generation. The strategy here is to use trio data to identify de novo mutations in children that are more common than expected. They use two large public data sets of trios with NDD or NDD+autism for loss of function mutations and the NDD for missense mutations. The three sets of genes identified from each approach are then compared to the results from a recent preprint on direct sequencing of mutations in sperm and also interpreted biologically with evidence that most identified genes are testis expressed and some of them have functions compatible with clonal expansion.

The study addresses an interesting problem and uses clever approaches to identify more mutations that are more common than expected. The partial concordance with the direct sequencing study (reference 20) suggests that some fraction of the findings are likely to be examples of clonal expansions.

However, I have several critical comments about the writing, the methodology, and the statistical/analysis choices made (1-8 below). I would need answers to these to be able to evaluate the strength of the evidence

1. Writing: The manuscript is very hard to read even for someone acquainted with the previous studies it rests on. There are excessive use of abbreviations and the trains-of-thoughts are hard to follow. Many times I have to take the authors' word for the findings reported.

One reason for it to be hard to follow is that very few numbers are being used throughout - for example, how many trios were analysed, how many mutations, what is the number of mutations needed for a significant LoF finding in each of the genes? I simply lack some feeling for the overall statistical power in the data. In Figure 1e, an o/e above 200, is that because you expect 0.01 mutations but observe two or expect one mutation but observe 200? It seems from the references that there are 10000 trios in the NDD set, in the exome this should correspond to a total of perhaps 10000 mutations for all genes where some smaller fraction are LoFs. What is the frequency distribution of mutations causing LoFs? Without such information, it is very hard to evaluate the evidence.

2. The roulette model. The roulette model has been shown in previous studies to capture much of the variation in mutation rates across the genome. However, here, it is very difficult to evaluate the risk of false positives that should be there even if roulette predicts (as claimed) 96% of the variance. I would need more evidence in the main Figures than Figure 1d to trust the threshold of $Z=3$ chosen. Also, why is synonymous outlier genes removed if more than 20% more mutations than expected (could be any number). Are the genes identified also higher in synonymous mutations than expected?

3. The data is by necessity from trios chosen to have affected children, either NDD or autism. This is a major confounder, as

also acknowledged by the authors. They try to use equation 1 as an upper limit, which is likely to be conservative, but this is also assuming complete penetrance and it is not clear exactly how this will work in the case where there are likely very few LoF mutations for any given gene. It is also very unclear how to disentangle true confounding (that CES actually cause NDD) from ascertainment and I do not think this has been done sufficiently (but unfortunately, I do not see how to solve it either)

4. I have similar comments to Figure 2 as Figure 1, why never show the number of mutations observed rather than the o/e ratios that can be very high just because e is very low? How is significance testing then even possible?

5. Most genes with evidence of CES are expressed in spermatogonia (or for set 3 in testis). Why these two different tests? I would always use spermatogonia. And what is the expected number - after all most genes are expressed at some level in testis. Again, very few numbers are provided and no tests of significance.

6. Figure 3a shows “minimal CES estimated from individual significant 20 missense variants” with a reference to the methods section. I fail to understand exactly how to interpret the numbers. At face value it looks like a single mutation can make a gene significant, but that is not true, correct?

7. The comparison to the nanoseq direct sequencing is interesting, but the overlap, even if significant, is not impressive. I have read the nanoseq preprint as well and since this is based on the direct sequencing of sperm in healthy donors, those results are much easier to interpret than the results here

8. The Methods section is fairly technical, which is fine, but again, it is difficult to read and it is even more difficult to relate to the main text, where it is often cited without any pointer as to where to find the supporting evidence in the methods section.

Referee #2

(Remarks to the Author)

This study by Seplyarskiy et al developed an approach to detect genes and mutations promoting clonal expansion in spermatogonia (CES) based on excess mutations observed in trios and/or in large population cohort, relative to a baseline mutation model. Applying this method to ~60,000 trios (90% with congenital disorders and 10% healthy controls) and population variation of ~800,000 unrelated individuals, the authors identified 15 genes with significant excess loss-of-function (LoF) mutations and 17 genes with excess missense mutations at specific sites. The authors hypothesized that these genes are enriched for mutations partially because they drive CES, although some of the detected enrichment signals come from pathogenicity of the genes combined with the phenotype ascertainment. Consistent with this hypothesis, these candidate genes are often expressed in spermatogonia, have substantial overlap with cancer genes that drive clonal expansion in somatic tissues, and overlap with CES driver genes identified with single-molecule sequencing of sperm samples.

Overall, I think this paper is very important, innovative, and timely, especially when accompanied with the discoveries of the independent study of Neville et al. The findings will reshape our understanding of germline mutation accumulation and modeling of baseline mutation rates. The clever ideas and careful execution of this study provide convincing evidence that many genes carry excess mutations in pedigrees and/or healthy populations, beyond what can be explained by regular mutation model or disease ascertainment, pointing to positive selection in the germline. These findings will no doubt have profound implications on future research, including both the study of germline mutagenesis and mutation-based study of disease risks and mechanisms.

That said, I do think this manuscript can be further strengthened in rigor and clarity, with minor additions in analysis and changes in writing and figure presentation. Although I agree with the authors that clonal expansion in spermatogonia (CES) is the most likely and parsimonious explanation for many genes in the short list, I am not convinced that some of the identified genes, in particular set 1 genes, fit the narrative, as they lack some of the expected signatures. In addition, the criteria for detecting set 2 genes (i.e., those with high LoF polymorphisms) seem arbitrary and may be improved. I am also not fully convinced that the positive selection on the identified genes happens only spermatogonia and would like to see a couple of additional analyses that demonstrate sex- and tissue- specificity. The manuscript is well written in general, but sometimes the rationale and/or evidence for certain arguments is missing or at least not clearly presented.

1. Credibility of set 1 genes: There is convincing biological evidence supporting that the four set 1 genes are pathogenic for NDD, but the evidence for CES comes primarily from violation of inequality (1). However, as the authors clearly stated, inequality (1) holds under the assumption of monogenic cause and no CES. Then the question is what will happen if the genes/mutations increase disease risk in an oligogenic or polygenic manner? Would that break of the inequality or make the inequality more conservative? Other lines of evidence of CES seem weak for set 1 genes: none has high LOEUF values or appears in COSMIC tier 1 cancer driver genes, only one overlaps with NanoSeq significant genes, and it is unclear whether set 1 genes are enriched for LoF DNMs in control trios (Fig 4b only shows the combined results of set 1 + set 2 genes). Did I miss something?

2. Criteria for detecting CES genes based on LoF polymorphisms: it is unclear why the authors chose to detect highly polymorphism genes using LOEUF with a hard-coded threshold of 0.5. LOEUF can be loosely interpreted as the upper bound of the confidence interval of O/E ratio. Designed as a conservative metric for detecting depletion of polymorphisms, LOEUF is likely not optimal for the authors' aim of identifying genes with excess polymorphisms. For example, short genes

tend to wider confidence intervals due to lower counts of observed LoF variants, rendering the short genes more likely to have high LOEUF and be mis-classified as “highly polymorphic”. Compared LOEUF, the point estimate (LOE) and the lower bound (LOELF) seem to be more logical options, as the formal is less biased, and latter more conserved. The arbitrary threshold 0.5 (previously used for detecting relaxed negative selection) puzzles me – can the authors provide some brief theoretical or empirical justification for this specific number? Lastly, most of the set 2 genes have $LOEUF < 1$, which technically still means significant depletion of LoF variants than neutral prediction. Is it accurate to describe them as “highly polymorphic” (as in the title of this section)?

3. Spermatogonia specificity: Suppose we accept that the candidate genes are under positive selection in the germline, in principle, selection can happen in the female germline or non-spermatogonia stages in males (e.g., progenitor germ cell proliferation, or even early embryogenesis, when cells are actively dividing). The various analyses of DNMs and gnomAD polymorphisms are largely agnostic of sex and tissue where the mutations arose. The overlap with CES genes identified by sperm sequencing provides strong support, but many genes are not in the overlap. Additional analyses may help strengthen the claim of positive selection in spermatogonia specifically.

1) The expression data in spermatogonia provide some support for CES, but do not exclude the possibility of selection in other tissues. In addition, it is known that many genes are transcribed at low level during spermatogenesis, potentially related to transcription noise during epigenetic reshaping from histone to protamine. Therefore, in addition to binary scoring of expression in spermatogonia, I would like to see quantitative expression profiles of the candidate genes in major cell types relevant to germline, such as early embryo, primary oocytes and male progenitor germ cells. The authors can then examine in which cells/stages, the candidate genes are expressed to the highest level, which may further support their CES hypothesis and perhaps generate new hypothesis (such as selection during early embryogenesis).

2) The parental origin can be determined for a substantial fraction of DNMs (15-30% depending on the sequencing read lengths). The claim of male germline-specific will be strengthened if there is nominally significant excess in paternal mutations. On the other hand, if the enrichment magnitude in kids born to older parents is significantly greater than that for younger parents, selection in early embryogenesis can be weakened. If some genes meet both predictions, CES hypothesis would be strongly supported. I understand that such sex- and age-stratified analysis may have limited power due to the much lower mutation counts; if this is the case, it would be helpful if the authors can perform some power simulations to find out the sample sizes needed for detecting such specific signals of CES (such large-scale studies are not available now but may become feasible in near future).

4. Suggestions for improving the result presentation:

1) It will be very helpful to include a table in one of the main figures that illustrate the expected signatures of different types of genes (in rows) in various datasets (in columns). For example, CES drivers are expected to have excess LoF mutations in ASD/NDD trios, control trios, sperm sequencing, and gnomAD cohort; in contrast, ASD/NDD pathogenic genes should have excess LoF mutations in ASD/NDD trios but not in normal trios or sperm sequencing, and perhaps slightly depleted in polymorphisms; in contrast, fitness-related genes would show depletion of LoF polymorphisms, and mild to no depletion in trios.

2) Also needed is a main table summarizing the mutation excess/depletion status in each cohort and additional information for the 32 candidate genes. If possible, the authors should also include their conclusion for the functional consequence of each gene (CES-driving? NDD pathogenic? Evolutionary constrained?)

3) For Fig 3a, an in-figure legend needs to be included to indicate the meaning of bold font and asterisks. Similar in-figure legends are needed in some other figure panels to help readers understand the figure content without having to look into legend texts.

4) In Fig 3d, it would make more sense to specify the sample size of COSMIC, corresponding to the observed mutation frequencies.

5) An explanation is needed to guide readers read Fig 4a and draw the conclusions of “CTNNB1 is significant ...” and “a set 2 gene, PTEN shows significant ascertainment ...”.

6) A brief explanation is needed for the claim “If CES drivers of set 1 do not cause NDD, the mutation rate should be increased by more than two orders of magnitude ...” on page 3.

Other minor points:

1. Although I understand how the authors reach the estimate of an average of 1.6 DNMs per site under extreme CES, I'm not sure how this number translates into 80%. Is this based on assumption of Poisson distribution? Would this number be much lower, if there is considerable site-level mutation rate variation?

2. The sentence “11 genes contribute the same amount of de novo variation as the average of 554 genes” on page 7 is very confusing. Do the authors mean that these 11 genes contribute a total amount of de novo variants equivalent to 554 times the average contribution of a randomly selected gene?

3. What does “In terms presented here” mean, on page 10?

4. One claim of this paper is that genes that drive CES may lead to false-positive findings in disease studies. This is a concern primarily when observed mutations in diseased cases are compared only to a baseline mutation model. However, many studies (for example of genetic risk of ASD) compare the mutation distribution between probands and healthy siblings, which would alleviate the concern. I hope the authors could explicitly specify the scenarios with this concern of false positives in the Introduction and suggest approaches to address this concern in Discussion.

Referee #3

(Remarks to the Author)

In this manuscript, Seplyarky et al sought to identify drivers of clonal expansion in spermatogonia (CES), which, in contrast to clonal expansions in somatic tissues, can be transmitted to the next generation. CES results in an increase in the de novo

mutation rate of the driver gene/ hotspot within the next generation and Seplyarksy et al leveraged this to identify CES drivers. Applying a systemic method to 54,715 trios (ascertained for rare conditions), 6065 control trios and gnomAD SNV data, they identified a total of 32 candidate CES genes, which they categorise into 3 different sets of genes, based on whether they were Loss of function(LoF) or Gain of function(GoF) and whether there was evidence of disease causality. Set 1 (4 genes) and set 2 (11 genes) were identified by looking for genes that had more Loss of function(LoF) mutations than could be explained by disease ascertainment. Set 1 genes showed evidence of disease causality, whereas set 2 genes did not. Set 3 (17 genes) were identified by looking for an excess of recurrent missense mutations, suggesting a gain of function(GoF) mechanism. Set 3 genes, like set 1, showed evidence of disease causality. They found that CES increases mutation rate ~16-fold for LoF genes and ~500-fold for pooled GoF sites. Recent Nanoseq sperm data identified 40 genes under position selection (based on enrichment for non-synonymous mutations) and, out of 32 candidate CES genes identified by Seplyarksy et al, 12 were significant in the Nanoseq data.

Overall, this work seems well conducted, the findings are interesting and the conclusions seem valid. By identifying genes that can drive CES with/without causing disease, this work is also of clinical importance. Indeed, despite the excess of LoF mutations in disease cohorts, only 5 of the 15 LoF CES driver candidates show clear evidence of disease causality.

I have a few questions related to the methods (and it might be helpful to include the answers to these within the methods):

- 1) What was the rationale for using missense variants as a proxy for GoF mutations?
- 2) The authors use the Roulette model to estimate the baseline mutation rate and they assess the accuracy of this by using de novo synonymous mutations in the largest trio dataset and show that the Roulette estimates are unbiased across more than 2 orders of magnitude of mutation rate. What about the impact of selection and hotspot effects on the Roulette model (for non-synonymous mutations)?
- 3) The authors mention that they use the Nei approximation – this typically relies on a defined effective population size. Surely the population size is difficult to estimate for sperm if they are undergoing continuous selection and expansion. How did the authors account for this?

I have a few questions related to the results (and it might be helpful to include the answers to some of these within the discussion):

- 1) There are some genes that were inferred to be CES drivers, but were not expressed in spermatogonia in single-cell RNAseq data (e.g. KCNA1 in set 2 and GRIN2B and PP2R5D in set 3). Do the authors have a hypothesis for why this is? Are they thought to be having their influence on clonal expansion via the RAS/MAPK pathway?
- 2) The authors found an average 16-fold increase in mutation rate for LoF CES genes and an average 524-fold increase in mutation rate at GoF positions. What do the authors think is the explanation for this? Could it be that the mutation rate is so much higher at GoF positions because the GoF mutation rates are at individual positions whereas the LoF mutation rates are across individual genes, where sites that are not under selection effectively dilute the genes mutation rate.
- 3) Figure 3a – I find it unclear how the #mutations in Nanoseq along the X-axis relates to the data shown on the plot/ the x-axis.
- 4) Regarding the CES driver genes that are also known cancer driver genes – do the authors think these could increase germline cancer risk, or are they simply genes that have a high selective advantage?
- 5) The authors state in the discussion that set 1 genes should have a substantial disease effect and moderate CES effect and that the Nanoseq sperm sequencing could be underpowered to identify CES for such genes. But how do the authors explain the CES drivers that were detected by Nanoseq sequencing, but not the author's method?

Version 1:

Reviewer comments:

Referee #1

(Remarks to the Author)

The authors have answered satisfactorily to all my comments. Importantly, they have incorporated these answers together with those of the other reviewers into a much-improved manuscript. It is now clear to me exactly how each analysis was performed and on which data. And the findings nicely complement those found by the nanoseq sequencing paper. I still find some of the arguments convoluted but that may just reflect the complexity of an analysis based on an ascertained set of trios and I do not have more suggestions for improving this (but I would appreciate if the authors have some since this could increase the readership of the paper).

My single concrete suggestion is that the authors could consider more in the discussion the evolutionary consequences of the process, like how much selection is needed for CES not to fix gain-of-function mutations, and an estimated total genetic load of the process. It could also be interesting to look up the prevalence of the identified CES mutations as a function of paternal age in the trios (maternal age as well), since CES should be exponentially increasing with age

Referee #2

(Remarks to the Author)

The authors have addressed the major concerns from the initial review, substantially enhancing the manuscript. All primary issues have been resolved, and the statistical tests employed appear appropriate, with the meaning of error bars in figures now clearly defined.

My remaining minor questions/suggestions pertain to wording and figure presentation:

1. To facilitate reader understanding, the meaning of variables V (presence of genetic variant) and D (ascertainment of disease) should be clearly indicated for Equation 1, before discussing other quantities like P(V) and P(D).
2. The term "cancerogenesis" on page 5 is less common; "carcinogenesis" is the preferred phrase.
3. In the last section of the discussion, "lethality" implies a dichotomous outcome and frequently requires the qualifier "partial." "Embryonic/prenatal deleteriousness" might be a more suitable substitution for "partial embryonic lethality."
4. Figures 1d and 4b are too small, hindering the readability of details, including error bars.
5. Figs 3b and 4b: the numbers below the bars are confusing. Variant classes should be directly below the bars, perhaps with the number of variants in parentheses beside the classes.
6. Fig 4d: the in-figure legend ("* significance in NDD") appears inconsistent with the caption's description ("genes with missenses significant in NDD are highlighted with blue shading").
7. Extended data figure 3: The y-axis label (LOELF) is inconsistent with the legend "Observed/Expected Upper bound Fraction."
8. Extended data figure 7: The label of the in-figure color scale legend seems incorrect. Should it be "power" instead of "(1-power)"? Power is expected to increase with mutation rate inflation factor κ and the number of variants, which is the opposite of what the figure currently implies. On a separate point, what does the current sample size lie in these plots? For $\kappa > 20$ (which is reasonable given Fig 4b), isn't there reasonable power at significance level of 0.01 with as few as 20 mutations?

Referee #3

(Remarks to the Author)

Thank you very much to the authors for responding to my comments and for their comprehensive revisions. I believe the manuscript is now substantially improved. Several previously unclear sections have been rewritten with greater clarity and are supported by helpful additional explanations. The explanation for their rationale for using missense variants as a proxy for GoF mutations was particularly helpful. The revised figures are also much improved, especially the replacement of Fig 3a with a more readable table (Fig 1e). I particularly note the more detailed discussion comparing the findings with Nanoseq, which I think strengthens the interpretation of the results.

I am happy that the author's have addressed all my comments/ concerns.

We thank the reviewers for their insightful comments and constructive feedback, which prompted us to improve the manuscript in several important ways. In response, we have substantially restructured the text and added necessary explanations and results that we believe make our study more convincing. Please find our detailed point-by-point responses below.

Reviewer 1

In this manuscript, the authors use a diverse set of approaches to identify genes of functional importance with more loss-of-function mutations (or missense mutations) than expected from an advanced prediction of site-specific mutation rates to identify genes that have the capacity to cause clonal expansion in spermatogonia of the male germline (and thus appear to have a higher mutation rate). We already know from a few well-studied cases that such clonal expansion can be very detrimental since the mutations causing them are more likely to cause syndromes in the next generation.

The strategy here is to use trio data to identify *de novo* mutations in children that are more common than expected. They use two large public data sets of trios with NDD or NDD+autism for loss of function mutations and the NDD for missense mutations. The three sets of genes identified from each approach are then compared to the results from a recent preprint on direct sequencing of mutations in sperm and also interpreted biologically with evidence that most identified genes are testis expressed and some of them have functions compatible with clonal expansion.

The study addresses an interesting problem and uses clever approaches to identify more mutations that are more common than expected. The partial concordance with the direct sequencing study (reference 20) suggests that some fraction of the findings are likely to be examples of clonal expansions.

However, I have several critical comments about the writing, the methodology, and the statistical/analysis choices made (1-8 below). I would need answers to these to be able to evaluate the strength of the evidence

1. Writing: The manuscript is very hard to read even for someone acquainted with the previous studies it rests on. There are excessive use of abbreviations and the trains-of-thoughts are hard to follow.

We have addressed the issue of clarity of the manuscript by substantially re-structuring the text. Specifically, we now introduce a more intuitive test for individual sites first, then generalize it to the LoF variation. Overall, we streamlined the logic of the paper and improved readability from the language perspective.

Many times I have to take the authors' word for the findings reported.

One reason for it to be hard to follow is that very few numbers are being used throughout - for example, how many trios were analysed, how many mutations, what is the number of mutations needed for a significant LoF finding in each of the genes? I simply lack some feeling for the overall statistical power in the data. In Figure 1e, an o/e above 200, is that because you expect 0.01 mutations but observe two or expect one mutation but observe 200? It seems from the references that there are 10000 trios in the NDD set, in the exome this should correspond to a total of perhaps 10000 mutations for all genes where some smaller fraction are LoFs. What is the frequency distribution of mutations causing LoFs? Without such information, it is very hard to evaluate the evidence.

We appreciate comments about lack of numbers and, as a consequence, lack of intuition about statistical properties of the effects. In response, we have made several improvements:

- We have supplemented Figures 1–3 with tables showing the counts of mutations, magnitudes of enrichments, and related values to provide a clearer and more quantitative view of the data.
- We have added a number of items with additional details: Extended Data Figure 1 (and see below here) with the histogram of *de novo* missense and LoF counts and Supplementary tables S4, S5, S7, S9, S11 with the numbers of observed *de novo* variants in all five processed *de novo* cohorts

Finally, we clarified in the main text that the number of considered trios is 31,058 in NDD and 16,877 in ASD. In the Main Text (“CES and loss-of-function polymorphism” section), we now have:

“...motivating us to increase power by merging *de novo* variation in cohorts ascertained by NDD (31,058 affected probands) and autism spectrum disorders (ASD, 16,877 affected probands)...”

Extended Data Figure 1 | Counts of *de novo* variants in the NDD cohort. (a) Numbers of *de novo* missense variants stratified by recurrence. Genes harboring variants occurring >10 times in the cohort are shown in blue. **(b)** Numbers of *de novo* loss-of-function variants aggregated by gene stratified by recurrence. **(c)** Scatter plot of observed vs. expected *de novo* loss-of-function

variant counts in the NDD cohort. LoF-1 set genes are shown in red; LoF-2 set genes are shown in purple. Upper bound of disease ascertainment in Eq (1) given by the lower bound of prevalence of 1% is shown as a dashed line.

2. The roulette model. The roulette model has been shown in previous studies to capture much of the variation in mutation rates across the genome. However, here, it is very difficult to evaluate the risk of false positives that should be there even if roulette predicts (as claimed) 96% of the variance. I would need more evidence in the main Figures than Figure 1d to trust the threshold of $Z=3$ chosen. Also, why is synonymous outlier genes removed if more than 20% more mutations than expected (could be any number). Are the genes identified also higher in synonymous mutations than expected?

In response to this comment, we have substantially updated the statistical procedure validating Roulette and the corresponding presentation in the text (Methods-5, first and third paragraphs in Results). Specifically, we replaced the population control with more appropriate *de novo* mutation tests.

We assess the possible Roulette errors on two levels: the per-site errors and the errors related to aggregation of Roulette rates to obtain the total mutation rate in the LoF class per gene. Consequently, we are using separate tests and separate quality controls of Roulette for the GoF and LoF sets. At the per-site level, we examine recurrence of synonymous variants in the NDD cohort and obtain the variance inflation relative to Poisson of 0.4%, which is incorporated in the Negative Binomial model.

A different source of Roulette error is a major concern when aggregating sites within a gene: non-independent errors may accumulate in the process of site aggregation (Methods-5), leading to Roulette over- or under-predicting mutation rate across the entire gene. To validate gene level expectations against an adequate null we used synonymous counts per gene in the NDD cohort, which indeed should follow Poisson distribution for a non-inflated model (we have actual number of counts per site instead of presence/absence like in the case of the gnomAD-based test). The gene-level tests on the *de novo* data include: 1) Poisson-Negative Binomial fit akin to the one previously done for individual sites, 2) Variance-based approach with explicit testing for the presence of detectable variation beyond Poisson and 3) The visual inspection of the distribution of per-gene synonymous *de novo* counts versus the Poisson expectation, which we present as a main Figure 2b. All of these tests confirm the absence of the gene-specific deviations from Roulette. We describe these tests in Methods-5 (“Control for Biases in Roulette”).

After careful considerations, we realized that the original population-based validation is based on incorrect assumptions rendering it overly conservative. This population test showed good concordance between expected count of polymorphic sites stratified by mutation rates and the one observed was based on slightly miscalibrated statistics. Specifically, the expected probability of mutation per site was calculated as the probability of

observing one or more mutations under Poisson with the rate given by Roulette. Indeed, the sum of these expectations should provide an unbiased mean. However, with the size of the current version of gnomAD, the presence of recurrent mutations cannot be disregarded, rendering Poisson a poor distributional approximation to the sum of mutation counts over genes. To give an intuition about the problem, for CpG >TpG sites the probability of an SNV is very close to 1, meaning that Poisson will have a substantial probability mass for values >1 and hence provide a poor approximation for probability of SNV, that by definition cannot exceed 1. We therefore excluded this test from the paper.

Figure 2b | Observed synonymous *de novo* mutation counts in the NDD cohort vs. the ones expected under the Poisson counts around Roulette estimates

3. The data is by necessity from trios chosen to have affected children, either NDD or autism. This is a major confounder, as also acknowledged by the authors. They try to use equation 1 as an upper limit, which is likely to be conservative, but this is also assuming complete penetrance

In response to this comment as well as related comments below, we altered the presentation in the main text and added a note in Supplementary Text-4 with a more formal treatment of the problem.

Equation (1) provides the lower bound of the CES effect on mutation rate under any possible disease ascertainment. In the absence of CES effect and under full penetrance, the inequality (1) becomes an equality. This means that in the absence of CES, mutation rate is upper-bounded by disease prevalence. In the presence of CES, the impact of CES on mutation rate equals the excess of observed mutation counts over the maximal ascertainment given by the inverse prevalence. Incomplete penetrance reduces the observed mutation counts in the disease cohort, rendering the effect of disease ascertainment smaller. For the purpose of finding CES drivers, this makes the test conservative.

For the set of LoF-2 genes, the effect of disease ascertainment is not important.

Also note that due to disease ascertainment increasing the observed counts, our approach yielding GoF and LoF-1 genes has enhanced power for CES drivers causing NDD.

and it is not clear exactly how this will work in the case where there are likely very few LoF mutations for any given gene.

As we present in the newly included tables (Figs. 2c and 3c), numbers of LoF mutations are generally in dozens, which provides a meaningful basis for inference. Since we used exome-wide FDR/Bonferroni, we believe that we have a good handle for the false positive results. We therefore believe the main challenge in our analyses is not excess false positives, but rather limited statistical power, particularly for short genes. Indeed, given the same CES effect and disease ascertainment, longer genes are more likely to reach statistical significance simply because they accumulate more mutations.

It is also very unclear how to disentangle true confounding (that CES actually cause NDD) from ascertainment and I do not think this has been done sufficiently (but unfortunately, I do not see how to solve it either)

We agree that the relationship between the CES and NDD etiology is a question of high interest and indeed a challenging issue. We have now expanded the Discussion section and added Table 1 to clarify how CES drivers might be distinguished between those that contribute to disease and those that do not or have minimal impact. For example, if we take into account independent clinical evidence, we find that genes such as *PTEN* and *CTNNA1* are well-established NDD risk genes, whereas others (e.g., *MIB1*, *TCF7L2*) appear to have little or no effect on disease risk.

In addition, we added guidance for future NDD gene mapping efforts clarifying how the CES effects should be handled: “*These [disease trio] studies may find evidence of the causal role in NDD not confounded by CES such as: i. transmission distortion and other types of familial segregation, ii. case-control analyses, iii. phenotypic similarity of mutation carriers and iv. functional assays in vitro and in vivo.*”

Table 1 | Different data modalities offer possibilities for distinguishing types of modulators of observed *de novo* mutation rates

	NDD	Clonal expansions			Partially penetrant embryonic lethality
		Spermatogonia	Early development	Oocyte progenitors	
de novo effect in cases	↑	↑	↑	↑	↓
de novo effect in controls	0	↑	↑	↑	↓
Effect in sperm sequencing	0	↑	↑	0	0
Segregation in population	↓	↑	↑	↑	↓
Other considerations					
Disease phenotype	yes	no	no	no	no
Somatic mosaicism	no	no	yes	no	no
Parental bias	baseline	paternal	50/50	maternal	baseline

4. I have similar comments to Figure 2 as Figure 1, why never show the number of mutations observed rather than the o/e ratios that can be very high just because e is very low? How is significance testing then even possible?

We have added tables accompanying the main text figures that show the actual number of mutations observed in each gene set (as mentioned above, dozens of LoF mutations per gene and multiple GoF mutations per site have been observed, see Figs. 1e, 2c and 3c).

Regarding significance testing, the methods to evaluate statistical significance on the discrete count data (including the extremely low count regime) are well established. Exact Poisson and Negative Binomial tests are commonly used on count data and are known to properly control type 1 error. The reason the exact tests are applicable here is that the expectation (the mutation rate predicted by Roulette) is considered known. This is in contrast to tests (such as Chi-square contingency test), where the expectation is estimated from the observed counts in data.

5. Most genes with evidence of CES are expressed in spermatogonia (or for set 3 in testis). Why these two different tests? I would always use spermatogonia. And what is the expected number - after all most genes are expressed at some level in testis. Again, very few numbers are provided and no tests of significance.

We thank the reviewer for highlighting this and we apologize for the confusion: there was a typo in the original figure annotations, we had actually used the same portion of the same dataset (spermatogonia expression data from Bush et al., 2024) in both cases.

In the updated manuscript, we have refined our analysis by using single-cell RNA-seq data from The Human Protein Atlas instead of the Bush et al. dataset. The Human Protein Atlas dataset is both more robust and widely accepted by the community. Specifically, we focus on the reported expression levels (TPMs) in spermatogonia. We have updated the Methods section accordingly, see Methods-1 (“Data”) and Methods-7 (“Expression and Gene Set Enrichment Analyses”).

We now include an Extended Data Figure 2 that shows the relative expression in spermatogonia of all CES driver candidates identified here. For this, we normalize the spermatogonia TPM of each gene by its maximum TPM across all tissues in the Human Protein Atlas, to provide a meaningful relative measure.

To address significance, we applied the Mann-Whitney U test comparing the spermatogonia expression of CES genes to that of non-CES genes, and we report the corresponding p-values in the main text.

6. Figure 3a shows “minimal CES estimated from individual significant 20 missense variants” with a reference to the methods section. I fail to understand exactly how to interpret the numbers. At face value it looks like a single mutation can make a gene significant, but that is not true, correct?

We agree that the original figure was confusing, and in the revised manuscript we have replaced it with a table that should provide a clearer interpretation. To clarify: it is never the case that a single mutation is sufficient to make a gene significant, nor is it theoretically possible under the realistic SNV mutation rates and the sizes of the samples used in our study. This is now evident from the mutation counts shown in Figure 1e.

In the GoF set, our analysis estimates the genome-wide significance of each missense site individually. We identify sites that show significantly more mutations than expected after correcting for ~50 million tests (reflecting all possible missense sites in the genome), and we further require the observed mutation count to exceed the expected mutation rate by a factor of 100 to ensure that results are not driven by disease ascertainment alone. We consider the approach conservative, as it tests the CES effect against the null assuming both lower bound on prevalence and full penetrance.

In the original manuscript, the “minimal CES effect” referred to the ratio between the actual number of mutations observed at a site and the minimal number of mutations that

would be required for that site to reach genome-wide significance under our stringent criteria.

7. The comparison to the nanoseq direct sequencing is interesting, but the overlap, even if significant, is not impressive. I have read the nanoseq preprint as well and since this is based on the direct sequencing of sperm in healthy donors, those results are much easier to interpret than the results here

In the revised manuscript, we have clarified several points to aid interpretation of the overlap to address this issue and also a related concern raised by Reviewer 2:

- In addition to showing the overlap, we now explicitly report in the Main Text that 17 out of 40 genes in our CES list are nominally significant in the NanoSeq dataset, supporting the relevance of our findings.
- The NanoSeq data, while offering a powerful approach to identify CES drivers, currently has some limitations that influence overlap. Notably, non-uniform coverage across the genome, which can reduce sensitivity for detecting CES drivers in certain genes.
- The number of observed mutations per gene in the NanoSeq dataset is generally low, which makes it challenging to achieve robust rankings of genes in any statistical setting. We note that our method also generates ranked lists of candidate CES genes based on discrete mutation counts, which should lead to incomplete overlap even if underlying CES effects are identical. To investigate this, we resampled LoF mutations from the NanoSeq dataset and estimated the overlap among $FDR < 0.2$ genes between two independent iterations of the resampling. Based on the results of this analysis, we hypothesize that the effect of inconsistent sorting on finite counts (also called the winner's curse) is the main reason for the incomplete ($\sim \frac{1}{3}$ on average) overlap between sampling replicates from NanoSeq (see Figure below).
- Importantly, our findings are supported by independent lines of evidence beyond the NanoSeq comparison: the signal present in the control cohort as well as cohorts ascertained by conditions other than ASD/NDD and enrichment in cancer driver and CES-relevant pathway. Together, these lines of evidence provide the strong support of our CES candidates.

Distribution for the fraction of FDR-significant genes in the overlap between two samples of mutations from the NanoSeq dataset. Red dashed line corresponds to overlap between FDR-significant genes from Neville et al, and LoF sets 1 and 2.

We also clarified this issue in the Main Text:

“Partial overlap between the lists could be attributed to multiple factors. First of all, datasets obtained by NanoSeq, have variable coverage among the genes, and indeed our CES candidates that are not significant in Neville et al, have lower coverage (Extended Data Figure 6). Second, for GoF-1 and LoF-1 the two approaches have differential statistical power. For the human genetics approach, the power is enhanced by the effect of CES drivers on NDD, whereas the power of the NanoSeq experiment is not boosted by the ascertainment. Finally, the power of both approaches is affected by sampling variance in the low mutation count data.

To demonstrate that the overlap is limited by power, we show that the residual effect exists in genes outside of the overlap. We calculated the observed-to-expected ratio for LoF mutations in the NanoSeq dataset for genes that did not attain genome-wide significance, but are included in our LoF sets (Fig 4b). We also measured the observed-to-expected ratio for genes outside the overlap using de novo variants in control trios (Fig 4b). In both cases, we observe significant enrichment of LoF mutations ($p < 0.01$, Poisson test).”

8. The Methods section is fairly technical, which is fine, but again, it is difficult to read and it is even more difficult to relate to the main text, where it is often cited without any pointer as to where to find the supporting evidence in the methods section.

We now have added numerical references to the subsection of the Methods.

Reviewer 2

This study by Seplyarskiy et al developed an approach to detect genes and mutations promoting clonal expansion in spermatogonia (CES) based on excess mutations observed in trios and/or in large population cohort, relative to a baseline mutation model. Applying this method to ~60,000 trios (90% with congenital disorders and 10% healthy controls) and population variation of ~800,000 unrelated individuals, the authors identified 15 genes with significant excess loss-of-function (LoF) mutations and 17 genes with excess missense mutations at specific sites. The authors hypothesized that these genes are enriched for mutations partially because they drive CES, although some of the detected enrichment signals come from pathogenicity of the genes combined with the phenotype ascertainment. Consistent with this hypothesis, these candidate genes are often expressed in spermatogonia, have substantial overlap with cancer genes that drive clonal expansion in somatic tissues, and overlap with CES driver genes identified with single-molecule sequencing of sperm samples.

Overall, I think this paper is very important, innovative, and timely, especially when accompanied with the discoveries of the independent study of Neville et al. The findings will reshape our understanding of germline mutation accumulation and modeling of baseline mutation rates. The clever ideas and careful execution of this study provide convincing evidence that many genes carry excess mutations in pedigrees and/or healthy populations, beyond what can be explained by regular mutation model or disease ascertainment, pointing to positive selection in the germline. These findings will no doubt have profound implications on future research, including both the study of germline mutagenesis and mutation-based study of disease risks and mechanisms.

That said, I do think this manuscript can be further strengthened in rigor and clarity, with minor additions in analysis and changes in writing and figure presentation. Although I agree with the authors that clonal expansion in spermatogonia (CES) is the most likely and parsimonious explanation for many genes in the short list, I am not convinced that some of the identified genes, in particular set 1 genes, fit the narrative, as they lack some of the expected signatures. In addition, the criteria for detecting set 2 genes (i.e., those with high LoF polymorphisms) seem arbitrary and may be improved. I am also not fully convinced that the positive selection on the identified genes happens only in spermatogonia and would like to see a couple of additional analyses that demonstrate sex- and tissue- specificity. The manuscript is well written in general, but sometimes the rationale and/or evidence for certain arguments is missing or at least not clearly presented.

1. Credibility of set 1 genes: There is convincing biological evidence supporting that the four set 1 genes are pathogenic for NDD, but the evidence for CES comes primarily from violation of inequality (1). However, as the authors clearly stated, inequality (1) holds under the assumption of monogenic cause and no CES. Then the

question is what will happen if the genes/mutations increase disease risk in an oligogenic or polygenic manner? Would that break of the inequality or make the inequality more conservative?

We now clarify in the Main Text that Equation (1) provides the exact upper bound for mutation counts under no CES, and this bound holds regardless of whether disease risk is conferred in a monogenic or polygenic (oligogenic) manner.

Importantly, in oligo- and polygenic scenarios where individual variants have lower penetrance, the inequality becomes more conservative — that is, the observed mutation excess required to violate the bound becomes larger. Therefore, the detection of CES using this criterion remains valid and conservative in the presence of polygenicity. To further support this point and address the related concerns raised by Reviewer 1, we have added a more formal treatment of this issue as the Supplementary Text 4.

Other lines of evidence of CES seem weak for set 1 genes: none has high LOEUF values or appears in COSMIC tier 1 cancer driver genes, only one overlaps with NanoSeq significant genes, and it is unclear whether set 1 genes are enriched for LoF DNMs in control trios (Fig 4b only shows the combined results of set 1 + set 2 genes). Did I miss something?

In the revised manuscript, we emphasize that all genes of the LoF-1 set (previously set 1) are strong biological candidates for CES. Specifically, this set includes 2 cancer genes (*ARID1B* and *CTNNB1*) and 1 gene with a recognized role in somatic clonal expansions (*DYRK1A*). As noted by the reviewer, one gene (*CTNNB1*) also overlaps with the significant findings from the NanoSeq study. Additionally, all genes are expressed in spermatogonia, and in the revised version we added Extended Data Figure 2 to emphasize this point.

To address the reviewer's concern, we have substantially revised the validation paragraph for the LoF-1 set in the Main Text. The updated text reads:

“Five genes display numbers of de novo LoF mutations in the NDD cohort significantly exceeding any plausible ascertainment by disease (by Eq. 1): PURA, ARID1B, CTNNB1, DYRK1A and FOXP1 (Fig. 2a, b). We call this list the LoF-1 set. Three of them, ARID1B, CTNNB1, DYRK1A, are involved in cancerogenesis or other clonal expansions (Fig 2b). Although PURA and FOXP1 have not been identified as drivers of somatic clonal expansions, their functions fit the profile of clonal expansion drivers. PURA is involved in replication and transcription control and FOXP1 is a transcription factor regulating early development.

The independent evidence of association with disease phenotypes supports the effect of LoF mutations on NDD for all five of these genes. Also, these genes are highly selectively constrained in the human population, consistent with their role in severe pediatric conditions.”

We have also added a table (Fig. 2, panel c) to the figure chain presented in the Main Text, summarizing the involvement of LoF-1 genes in cancer and other somatic clonal expansions.

2. Criteria for detecting CES genes based on LoF polymorphisms: it is unclear why the authors chose to detect highly polymorphism genes using LOEUF with a hard-coded threshold of 0.5. LOEUF can be loosely interpreted as the upper bound of the confidence interval of O/E ratio. Designed as a conservative metric for detecting depletion of polymorphisms, LOEUF is likely not optimal for the authors' aim of identifying genes with excess polymorphisms. For example, short genes tend to wider confidence intervals due to lower counts of observed LoF variants, rendering the short genes more likely to have high LOEUF and be mis-classified as "highly polymorphic". Compared LOEUF, the point estimate (LOE) and the lower bound (LOELF) seem to be more logical options, as the former is less biased, and latter more conserved. The arbitrary threshold 0.5 (previously used for detecting relaxed negative selection) puzzles me – can the authors provide some brief theoretical or empirical justification for this specific number? Lastly, most of the set 2 genes have $LOEUF < 1$, which technically still means significant depletion of LoF variants than neutral prediction. Is it accurate to describe them as "highly polymorphic" (as in the title of this section)?

We originally selected LOEUF because it is a more familiar and widely used point of reference in the human genetics community for summarizing gene-level constraint. For validation, we used additional tests based on *de novo* mutations in controls, other disease cohorts and population genetic variation. We agree with the reviewers that the use of LOEUF with a specific threshold is arbitrary. We investigated robustness of the inference with respect to the threshold and the method of estimating selective constraint. We now show on Extended Data Figure 4 that the results are highly robust with respect to the LOEUF threshold. We also now analyzed the data using adjusted thresholds (to include all the genes from LoF-set 2, previously set 2) of LOE and LOELF (shown in the new Extended Data Figure 3). The results are highly concordant with the original findings, albeit the LOE analysis adds two additional genes near the threshold, one of which (*ZMYM2*) is a plausible CES candidate and is nominally significant in the NanoSeq dataset. We also provide the analysis that incorporates *GeneBayes* as a different constraint metric (see Methods-8).

We agree with the Reviewers on the wording and do not refer to the LoF-2 genes (formerly set 2) as "highly polymorphic".

To address the reviewer's concern and investigate whether gene length is a substantial confounder with respect to LOEUF and its effect on our results, we made two observations. We compared distributions of LoF mutation targets between ASD/NDD significant genes with $LOEUF < 0.5$ and ASD/NDD significant genes $LOEUF > 0.5$. We have not identified a statistically significant difference (KS p-value 0.17). More convincingly, the distribution of LOEUF values for ASD/NDD significant genes appears bimodal (Figure 3a) in sharp contrast to the unimodal distribution of their mutation targets.

Extended Data Figure 3 | Stability of LoF-2 set with respect to the metric of loss-of-function constraint. (a) Observed-to-expected variant count ratio (o/e) for *de novo* LoFs in genes with FDR < 0.1 in the neurodevelopmental disorder cohort (NDD) merged with the autism spectrum disorder (ASD) cohort plotted against the Loss-of-function Observed/Expected Upper-bound Fraction (LOELF) scores. The dashed violet line indicates the minimal LOELF value across LoF-2 genes of 0.23. LoF-2 genes are shown in violet, LoF-1 genes are shown in red, genes above the chosen LOELF threshold but not included in the LoF-2 set (*SHANK3* and *ZMYM2*) are shown in black. (b) Same as in (a), but for the Loss-of-function Observed/Expected (LOE) metric. The upper bound for LOE (shown as violet dashed line) is 0.355.

Extended Data Figure 4 | Ratio of observed-to-expected counts of LoF *de novo* mutations in a cohort of control trios for LoF-2 set genes. The ratio is shown as a function of the LOEUF threshold: we aggregate all genes with LOEUF values lower than the value indicated on the x-axis and calculate the cumulative observed-to-expected ratio. The shaded grey area represents the 95% confidence interval obtained by permuting the LOEUF labels.

3. Spermatogonia specificity: Suppose we accept that the candidate genes are under positive selection in the germline, in principle, selection can happen in the female germline or non-spermatogonia stages in males (e.g., progenitor germ cell proliferation, or even early embryogenesis, when cells are actively dividing). The various analyses of DNMs and gnomAD polymorphisms are largely agnostic of sex and tissue where the mutations arose.

We thank the reviewer for bringing up this important point. In the revised manuscript, we have added a dedicated paragraph in the Discussion that explicitly addresses this issue. Briefly, although there is a strong argument in favor of the spermatogonia origin of most of the expansions, we agree that our analysis (and even direct sperm sequencing) cannot rule out the hypothesis of clonal expansions in early development. In the updated text, we have: *“Although here we interpret the elevation of mutation rate relative to the baseline as the effect of CES, the mutation rate increase may be due to clonal expansions in early development or in oocyte progenitors. Formally, our approach does not pinpoint the source of the clonal expansions. Still, substantial overlap (given statistical limitations) with sperm sequencing data supports the CES hypothesis. No mutations reported by Neville et al. have appreciable variant allele frequency in sperm²¹, ruling out early developmental origin. Historically, CES is the only reported clonal mechanism responsible for the elevation of germline mutation rate.*

In this study we have shown that the phenomenon of CES might be of unexpectedly great importance for future studies in the fields of genetics of rare disease, population genetics, cancer biology and in the emerging field of clonal evolution in somatic tissues.”

We also added a new table to the Main Text (Table 1) summarizing how different data modalities could help to differentiate between alternative sources of elevated observed mutation rates:

Table 1 | Different data modalities offer possibilities for distinguishing types of modulators of observed *de novo* mutation rates

	NDD	Clonal expansions			Partially penetrant embryonic lethality
		Spermatogonia	Early development	Oocyte progenitors	
de novo effect in cases	↑	↑	↑	↑	↓
de novo effect in controls	0	↑	↑	↑	↓
Effect in sperm sequencing	0	↑	↑	0	0
Segregation in population	↓	↑	↑	↑	↓
Other considerations					
Disease phenotype	yes	no	no	no	no
Somatic mosaicism	no	no	yes	no	no
Parental bias	baseline	paternal	50/50	maternal	baseline

We believe that this addition highlights the potential and limitations of current data types in disentangling the origin of elevated mutation rates.

The overlap with CES genes identified by sperm sequencing provides strong support, but many genes are not in the overlap. Additional analyses may help strengthen the claim of positive selection in spermatogonia specifically.

In response to the Reviewer’s concern and also to address a related concern raised by Reviewer 1, we have clarified this point in the revised manuscript.

- In addition to showing the overlap, we now explicitly report in the Main Text that 17 out of 40 genes in our CES list are nominally significant in the NanoSeq dataset, supporting the relevance of our findings.
- The NanoSeq data, while offering a powerful approach to identify CES drivers, currently has some limitations that influence overlap. Notably, non-uniform coverage across the genome, which can reduce sensitivity for detecting CES drivers in certain genes.
- The number of observed mutations per gene in the NanoSeq dataset is generally low, which makes it challenging to achieve robust rankings of genes in any statistical setting. We note that our method also generates ranked lists of candidate CES genes based on discrete mutation counts, which should lead to incomplete overlap even if underlying CES effects are identical. To investigate this, we resampled LoF mutations from the NanoSeq dataset and estimated the overlap among FDR < 0.2 genes between two independent iterations of the resampling. Based on the results of this analysis, we hypothesize that the effect of inconsistent sorting on finite counts (also called the winner's curse) is the main reason for the incomplete ($\sim \frac{1}{3}$ on average) overlap between sampling replicates from NanoSeq (see Figure below).
- Importantly, our findings are supported by independent lines of evidence beyond the NanoSeq comparison: the signal present in the control cohort as well as cohorts ascertained by conditions other than ASD/NDD and enrichment in cancer driver and CES-relevant pathway. Together, these lines of evidence provide the strong support of our CES candidates.

Distribution for the fraction of FDR-significant genes in the overlap between two samples of mutations from the NanoSeq dataset. Red dashed line corresponds to overlap between FDR-significant genes from Neville et al, and LoF sets 1 and 2.

We also clarified this issue in the Main Text:

“Partial overlap between the lists could be attributed to multiple factors. First of all, datasets obtained by NanoSeq, have variable coverage among the genes, and indeed our CES candidates that are not significant in Neville et al, have lower coverage (Extended Data Figure 6). Second, for GoF-1 and LoF-1 the two approaches have differential statistical power. For the human genetics approach, the power is enhanced by the effect of CES drivers on NDD, whereas the power of the NanoSeq experiment is not boosted by the ascertainment. Finally, the power of both approaches is affected by sampling variance in the low mutation count data.

To demonstrate that the overlap is limited by power, we show that the residual effect exists in genes outside of the overlap. We calculated the observed-to-expected ratio for LoF mutations in the NanoSeq dataset for genes that did not attain genome-wide significance, but are included in our LoF sets (Fig 4b). We also measured the observed-to-expected ratio for genes outside the overlap using de novo variants in control trios (Fig 4b). In both cases, we observe significant enrichment of LoF mutations ($p < 0.01$, Poisson test).”

As the reviewer rightly notes, further evidence for CES as the underlying mechanism could come from the paternal origin of mutations and their correlation with paternal age. We have contacted the authors of the Kaplanis et al., 2020 study (where *de novo* NDD mutation dataset was initially published) about the availability of paternal age and/or parent of origin data. As the authors have told us, this data is currently next to impossible to obtain. Still, to understand limitations of this approach, we conducted a power analysis showing that detecting CES based solely on an excess of *de novo* mutations of paternal origin would require substantially larger sample sizes than currently available for most genes. We have added an Extended Data Figure 7 to illustrate this. Also note that due to phasing limitations in short read data usually only for ~30% of *de novo* mutations, the parent of origin may be inferred.

Extended Data Figure 7 | Power analysis of paternal transmission. Statistical power (i.e., the probability of correctly detecting a signal when it exists) of the Binomial test for paternal overtransmission relative to the baseline of 0.75 is shown across the range of CES-related mutation rate inflations κ and counts of observed variants. Results are presented for three significance levels: 0.05, 0.01, and 0.001.

Finally, as noted above, the revised manuscript includes an expanded Discussion section outlining other plausible explanations for the observed mutation excesses (including clonal expansions in early development or oocyte progenitors) and explicitly discussing the value of phasing data for future studies:

“...many of the recovered CES driver candidates ... have no strong additional supporting evidence for NDD meaning that they could be false-positives in studies of gene-disease association that rely solely on de novo enrichment. However, these studies may find evidence of the causal role in NDD not confounded by CES such as: i. transmission distortion and other types of familial segregation (Extended Data Figure 7), ii. case-control analyses, iii. phenotypic similarity of mutation carriers and iv. functional assays in vitro and in vivo.”

1) The expression data in spermatogonia provide some support for CES, but do not exclude the possibility of selection in other tissues. In addition, it is known that many genes are transcribed at low level during spermatogenesis, potentially related to transcription noise during epigenetic reshaping from histone to protamine. Therefore, in addition to binary scoring of expression in spermatogonia, I would like to see quantitative expression profiles of the candidate genes in major cell types relevant to germline, such as early embryo, primary oocytes and male progenitor germ cells. The authors can then examine in which cells/stages, the candidate genes are expressed to the highest level, which may further support their CES hypothesis and perhaps generate new hypothesis (such as selection during early embryogenesis).

We have substantially expanded our analysis of expression patterns to better address the possibility of selection in tissues and stages beyond spermatogonia. Specifically, for the revised manuscript, we replaced the Bush et al. 2023 dataset of single-cell sperm sequencing with a more comprehensive resource from The Human Protein Atlas, which includes normalized quantitative expression data across a broad range of tissues and cell types. We now provide quantitative expression profiles (TPM values) for our candidate genes in spermatogonia and oogonia, normalized by the maximal expression of each gene across tissues (Extended Data Figure 2). This allows direct comparison of relative expression in these key germline progenitor cell types.

We agree with the Reviewer that expression in oocyte progenitors would provide valuable additional insight. However, high-quality data for human oocyte progenitors are unfortunately lacking. Existing work (e.g., Wagner et al. 2020, *Single-cell analysis of human ovarian cortex identifies distinct cell populations but no oogonial stem cells*) argues that expression profiles in oocytes do not align with those observed in oogonia, and highlights the absence of robust data on oocyte progenitor stages.

To further explore expression during early development, we have incorporated data from Xu et al. 2023 (*A single-cell transcriptome atlas profiles early organogenesis in human embryos*). We now provide a Supplementary Figure showing the expression of candidate genes across clusters identified in this study. This complements our germline analysis by offering a broader developmental context. We note, however, that the embryos in the Xu et

al. dataset represent stages at 4–6 weeks post-conception. Since clonal expansions arising in early development would likely occur prior to these stages, these data may not fully capture relevant expression patterns for detecting selection at very early embryonic stages. Unfortunately, to our knowledge, single-cell transcriptomic datasets covering earlier human embryonic stages are not currently available.

2) The parental origin can be determined for a substantial fraction of DNMs (15-30% depending on the sequencing read lengths). The claim of male germline-specific will be strengthened if there is nominally significant excess in paternal mutations. On the other hand, if the enrichment magnitude in kids born to older parents is significantly greater than that for younger parents, selection in early embryogenesis can be weakened. If some genes meet both predictions, CES hypothesis would be strongly supported. I understand that such sex- and age-stratified analysis may have limited power due to the much lower mutation counts; if this is the case, it would be helpful if the authors can perform some power simulations to find out the sample sizes needed for detecting such specific signals of CES (such large-scale studies are not available now but may become feasible in near future).

We fully agree with the Reviewer that parental origin and parental age analyses represent valuable orthogonal strategies for testing the CES hypothesis.

Unfortunately, we were unable to obtain access to raw sequencing data or to phased *de novo* mutations from the available cohorts, despite contacting the authors of the relevant disease studies. As the Reviewer suggests, such analyses would indeed provide complementary information to sperm sequencing: excess paternal-origin mutations and age-dependent enrichment would support CES, whereas selection during early embryogenesis would similarly produce functional mutation excess but without paternal bias.

As noted in our answer to a related concern above, in line with the Reviewer's recommendation, we have now performed a power analysis to estimate the sample sizes that would be required to detect a CES effect through parental-origin or age-stratified analyses. We present these results in the Extended Data Figure 7. We believe these results may be informative for designing future large-scale studies of CES.

4. Suggestions for improving the result presentation:

1) It will be very helpful to include a table in one of the main figures that illustrate the expected signatures of different types of genes (in rows) in various datasets (in columns). For example, CES drivers are expected to have excess LoF mutations in ASD/NDD trios, control trios, sperm sequencing, and gnomAD cohort; in contrast, ASD/NDD pathogenic genes should have excess LoF mutations in ASD/NDD trios but not in normal trios or sperm sequencing, and perhaps slightly depleted in polymorphisms; in contrast, fitness-related genes would show depletion of LoF polymorphisms, and mild to no depletion in trios.

As we note above, we fully agree with the Reviewer that such a summary table can greatly help readers interpret our findings across data modalities. In response, we have created Table 1 (in the Main Text) that illustrates the expected signatures for different classes of genes — CES drivers, ASD/NDD pathogenic genes, and others — across the key datasets analyzed in this study (ASD/NDD trios, control trios, sperm sequencing, and polymorphism data from gnomAD). In addition, we have added a paragraph to the Discussion to guide the interpretation of this table (see above). We believe that this addition helps clarify the interpretations and provides some guidance for future efforts.

2) Also needed is a main table summarizing the mutation excess/depletion status in each cohort and additional information for the 32 candidate genes. If possible, the authors should also include their conclusion for the functional consequence of each gene (CES-driving? NDD pathogenic? Evolutionary constrained?)

As we noted above and also in response to Reviewer 1's related concerns, we now include detailed tables in Figures 1–3 of the revised manuscript that summarize the mutation excess for each of the candidate genes across the relevant datasets. These tables also provide magnitudes of the enrichment and the overlap with previously known somatic clonal expansion drivers and disease genes.

In addition, as described in our responses above, we now provide Table 1, which summarizes the expected signatures for different functional classes of genes across these data modalities. This table is designed to help readers interpret our findings and assess the likely functional consequence of each gene based on the combined evidence. We hope that these additions significantly improve clarity and accessibility of the results.

3) For Fig 3a, an in-figure legend needs to be included to indicate the meaning of bold font and asterisks. Similar in-figure legends are needed in some other figure panels to help readers understand the figure content without having to look into legend texts.

In the revised manuscript, we removed Figure 3a in its previous form; it has been replaced with a more readable table (now Fig. 1e) that presents the relevant information clearly. In addition, all figures and tables now include appropriate in-figure legends, so that readers can understand the panels without referring to the Methods section or to the figure legends in the Main Text.

4) In Fig 3d, it would make more sense to specify the sample size of COSMIC, corresponding to the observed mutation frequencies.

In the revised manuscript this panel has been removed because it diverted the main story.

5) An explanation is needed to guide readers read Fig 4a and draw the conclusions of “CTNNB1 is significant ...” and “a set 2 gene, PTEN shows significant ascertainment ...”.

To improve readability, we have removed these parts from the Main Text. However, the necessary context for interpreting the effects of *CTNNB1* and *PTEN* on CES and NDD is now provided in Table 1 and related discussion.

6) A brief explanation is needed for the claim “If CES drivers of set 1 do not cause NDD, the mutation rate should be increased by more than two orders of magnitude ...” on page 3.

We have substituted this part with a more readable sentence in the “CES and loss-of-function polymorphism” section:

“The approach outlined above [i.e. the one based on violation of Eq. 1] is not suited well to identify such genes [i.e. CES drivers with weak or no effect on NDD], because it requires the counts of de novo mutations in the NDD cohort to exceed the expectation by at least 100 fold (inverse prevalence of NDD)...”.

Other minor points:

1. Although I understand how the authors reach the estimate of an average of 1.6 DNMs per site under extreme CES, I’m not sure how this number translates into 80%. Is this based on assumption of Poisson distribution? Would this number be much lower, if there is considerable site-level mutation rate variation?

We have removed this part in the new structure of the manuscript to improve clarity and focus.

2. The sentence “11 genes contribute the same amount of *de novo* variation as the average of 554 genes” on page 7 is very confusing. Do the authors mean that these 11 genes contribute a total amount of *de novo* variants equivalent to 554 times the average contribution of a randomly selected gene?

We have substituted this sentence with “*Finally, using the cohort of healthy controls, we additionally estimated that just 15 LoF-2 genes [previously set 2] explain 3.8% of de novo LoF variation.*”

3. What does “In terms presented here” mean, on page 10?

We have removed the phrase “*In terms presented here*” from the sentence to improve clarity. The sentence now reads:

“Direct sequencing of sperm is effectively equivalent to assaying paternal de novo mutations in a large cohort of randomly selected trios in all genes barring those which are not compatible with live births...”

4. One claim of this paper is that genes that drive CES may lead to false-positive findings in disease studies. This is a concern primarily when observed mutations in diseased cases are compared only to a baseline mutation model. However, many studies (for example of genetic risk of ASD) compare the mutation distribution between probands and healthy siblings, which would alleviate the concern. I hope the authors could explicitly specify the scenarios with this concern of false positives in the Introduction and suggest approaches to address this concern in Discussion.

We have added explicit clarification in the *Discussion* to specify the scenarios where CES could lead to false-positive associations and to highlight approaches that can help mitigate this concern. The revised text now states:

“...many of the recovered CES driver candidates, especially the ones from the LoF-2 set [previously set 2], e.g. MIB1 or TCF7L2, have no strong additional supporting evidence for NDD meaning that they could be false-positives in studies of gene-disease association that rely solely on de novo enrichment. However, these studies may find evidence of the causal role in NDD not confounded by CES such as: i. transmission distortion and other types of familial segregation, ii. case-control analyses, iii. phenotypic similarity of mutation carriers and iv. functional assays in vitro and in vivo.”

Reviewer 3

In this manuscript, Seplyarky et al sought to identify drivers of clonal expansion in spermatogonia (CES), which, in contrast to clonal expansions in somatic tissues, can be transmitted to the next generation. CES results in an increase in the *de novo* mutation rate of the driver gene/ hotspot within the next generation and Seplyarky et al leveraged this to identify CES drivers. Applying a systemic method to 54,715 trios (ascertained for rare conditions), 6065 control trios and gnomAD SNV data, they identified a total of 32 candidate CES genes, which they categorise into 3 different sets of genes, based on whether they were Loss of function (LoF) or Gain of function (GoF) and whether there was evidence of disease causality. Set 1 (4 genes) and set 2 (11 genes) were identified by looking for genes that had more Loss of function (LoF) mutations than could be explained by disease ascertainment. Set 1 genes showed evidence of disease causality, whereas set 2 genes did not. Set 3 (17 genes) were identified by looking for an excess of recurrent missense mutations, suggesting a gain of function (GoF) mechanism. Set 3 genes, like set 1, showed evidence of disease causality. They found that CES increases mutation rate ~16-fold for LoF genes and ~500-fold for pooled GoF sites. Recent Nanoseq sperm data identified 40 genes under position selection (based on enrichment for non-synonymous mutations) and,

out of 32 candidate CES genes identified by Seplyarky et al, 12 were significant in the Nanoseq data.

Overall, this work seems well conducted, the findings are interesting and the conclusions seem valid. By identifying genes that can drive CES with/without causing disease, this work is also of clinical importance. Indeed, despite the excess of LoF mutations in disease cohorts, only 5 of the 15 LoF CES driver candidates show clear evidence of disease causality.

I have a few questions related to the methods (and it might be helpful to include the answers to these within the methods):

1) What was the rationale for using missense variants as a proxy for GoF mutations?

We thank the reviewer for this thoughtful question. In brief, gain-of-function (GoF) mutations, unlike loss-of-function (LoF) mutations, are expected to cluster at specific sites within a gene rather than being scattered across the gene. Known CES mutations to date have all been isolated, individual missense variants acting through GoF mechanisms. Therefore, our method is tailored to detect such concentrated signals: for missense variants, we apply a per-site test (Eq. 1), which is sensitive to mutation enrichment at specific positions but has limited power if the signal is broadly distributed.

We have clarified this point in the revised manuscript by adding:

“All CES genes discovered to date act through the gain-of-function (GoF) mechanism. Known CES mutations are isolated individual missense variants. Given that the statistical signal expected for GoF mutations is highly concentrated in specific positions, we first apply Eq. (1) to individual missense de novo mutations in the NDD cohort.”

2) The authors use the Roulette model to estimate the baseline mutation rate and they assess the accuracy of this by using *de novo* synonymous mutations in the largest trio dataset and show that the Roulette estimates are unbiased across more than 2 orders of magnitude of mutation rate. What about the impact of selection and hotspot effects on the Roulette model (for non-synonymous mutations)?

When training the Roulette model, we explicitly excluded exonic sequences to prevent the model from learning features influenced by selection on protein function. Additionally, we trained Roulette using rare single nucleotide variants (SNVs), which are less affected by natural selection. As a result, Roulette’s mutation rate estimates are expected to reflect primarily biochemical and genomic properties of DNA—such as sequence context and replication direction—rather than functional effects like loss-of-function (LoF) or missense impacts.

To validate this, we tested Roulette’s predictions on synonymous sites, which are generally neutral with respect to any phenotype, confirming that the model provides unbiased estimates across more than two orders of magnitude in mutation rates. It is

unlikely that Roulette would accurately capture mutation rates for synonymous sites but fail to do so for LoF mutations, which are subject to similar biochemical mutation processes.

Moreover, we observe good concordance between observed and expected LoF mutation counts in the control cohort (291 observed vs. 321 expected; Poisson $p = 0.08$), further supporting the model's accuracy for functional categories. We want to mention that slightly lower than expected number of LoF mutations, despite being insignificant, could be driven by LoFs leading to embryonic lethality and ascertainment against disease causing LoFs in healthy cohorts.

We have improved the descriptions of our validation procedures in the Main text and in the Methods for clarity and added a number of checks on the gene level following this comment and similar comments made by Reviewer 1.

3) The authors mention that they use the Nei approximation – this typically relies on a defined effective population size. Surely the population size is difficult to estimate for sperm if they are undergoing continuous selection and expansion. How did the authors account for this?

We believe there may be some confusion regarding the use of the Nei approximation. The Nei approximation was applied specifically to estimate selection and the effect of CES using loss-of-function (LoF) single nucleotide variants from the gnomAD dataset. In this context, the effective population size refers to the human population rather than the sperm cell population.

To improve clarity, we have re-written relevant sections of the manuscript to better explain this distinction and ensure readers can follow the rationale, please see the revised “*CES and loss-of-function polymorphism*” paragraph in the Main Text as well as “*9. Variants in LoF-2 Genes Segregate in the Population Due to Elevated Mutation Rate*” in the Methods.

I have a few questions related to the results (and it might be helpful to include the answers to some of these within the discussion):

1) There are some genes that were inferred to be CES drivers, but were not expressed in spermatogonia in single-cell RNAseq data (e.g. *KCNA1* in set 2 and *GRIN2B* and *PPP2R5D* in set 3). Do the authors have a hypothesis for why this is? Are they thought to be having their influence on clonal expansion via the RAS/MAPK pathway?

In response to this concern (as well as related points raised above), we updated our expression analyses by switching to the more comprehensive single-cell RNA-seq dataset from The Human Protein Atlas to improve resolution. This allowed us to exclude *PPP2R5D* from the list of potential false positives due to lack of expression in spermatogonia; however, *GRIN2B* and *KCNA1* still showed no detectable expression.

The most parsimonious explanation for *KCNA1* (LoF-2 set, formerly set 2) and *GRIN2B* (GoF set, formerly set 3) is that they represent false positives, consistent with our use of a 10% FDR threshold for genome-wide significance, which allows for some false discoveries. We now explicitly mention this in the Main Text:

- “All but one of these genes are expressed in spermatogonia ($p=6.5\times 10^{-7}$). The exception, *GRIN2B*, was thus excluded from the GoF set (Extended Data Figure 2) as a potential false positive.”
- “Second, all LoF-2 genes [previously set 2] barring *KCNA1* are expressed in spermatogonia ... *KCNA1* was flagged as a potential false-positive finding.”

While we do not exclude the possibility that either gene might influence clonal expansion via pathways such as RAS/MAPK, we currently lack direct evidence supporting such mechanisms.

2) The authors found an average 16-fold increase in mutation rate for LoF CES genes and an average 524-fold increase in mutation rate at GoF positions. What do the authors think is the explanation for this? Could it be that the mutation rate is so much higher at GoF positions because the GoF mutation rates are at individual positions whereas the LoF mutation rates are across individual genes, where sites that are not under selection effectively dilute the genes mutation rate.

We can think of two main explanations for the observed discrepancy between the average mutation rate elevations in LoF CES genes (~16-fold) versus GoF positions (~524-fold).

First, spermatogonia cells are diploid, and the variants discussed here are likely heterozygous. LoF mutations typically have weaker effects in heterozygous state because the intact second copy of the gene can partially compensate. In contrast, GoF mutations often act dominantly at the cellular level, potentially driving stronger clonal expansions.

Second, from a population genetics perspective, the large number of potential LoF CES driver mutations within a gene contrasts with the relatively few GoF CES driver positions. LoF mutations within a gene generally share similar functional consequences, so many sites can contribute to CES. However, a 500-fold increase in mutation rate for highly penetrant, large NDD genes (e.g., *ARID1B* or *ARID1A*) would imply that approximately 0.5% of newborns carry severe NDD-causing mutations solely due to CES at a single gene. Such a high pathogenic burden would induce strong negative selection on hypermutability of these LoF genes. Conversely, GoF mutations affect only a few specific positions, so even a large mutation rate increase at these sites imposes a smaller overall selective burden on the population.

In response to the Reviewer’s relevant question, we have summarized these ideas in the Discussion:

“There is a discrepancy in the average elevations of mutation rate due to CES between GoF and LoF mutations (16-fold vs 524-fold). We propose two possible explanations. First, spermatogonia cells are diploid, and the variants discussed here are likely heterozygous. LoF mutations are often partially recessive with moderate effects in heterozygotes. In

contrast, GoF mutations are usually dominant. The second explanation comes from population genetics. We observe very limited numbers of GoF CES drivers in a single gene (Fig 1e). On the other hand, since all LoF mutations within a gene have identical functional consequences, the number of potential CES driver mutations is large. As CES drivers are frequently involved in severe diseases, 500-fold elevation in mutation rate might generate a substantial rate of de novo pathogenic mutations, e.g. for ARID1B, about 0.5% of all individuals would be born with NDD if the CES effect was so high. Such pathogenic burden created by LoF mutations in a single gene will trigger efficient negative selection on the CES effect.

3) Figure 3a – I find it unclear how the #mutations in Nanoseq along the X-axis relates to the data shown on the plot/ the x-axis.

In response to this concern, as well as related feedback from other reviewers about Figure 3a, we have removed the figure entirely and replaced it with a more readable table now on Fig 1e. We believe this change improves clarity and would help the readers to better understand the presented results and data. We are summarising overlap between GoF-set (previously set 3) and NanoSeq data by calculating that in NanoSeq the average rate of mutations across GoF sites is 524-fold higher.

4) Regarding the CES driver genes that are also known cancer driver genes – do the authors think these could increase germline cancer risk, or are they simply genes that have a high selective advantage?

Intriguingly, although there is substantial overlap between our list of CES genes and COSMIC Tier 1 cancer drivers, the most prominent drivers of testicular tumors—KIT, KRAS, and HRAS—are notably absent from our candidate CES genes (see Shen et al., 2018 *Integrated Molecular Characterization of Testicular Germ Cell Tumors*). This suggests that while these genes may confer a selective advantage in the germline, they do not necessarily increase germline cancer risk in a straightforward manner. Still, there is non-trivial overlap with drivers of testicular tumors, so we added an observation to Discussion: *“As expected for genes involved in clonal expansions, 12 out of 40 genes identified here are COSMIC census cancer drivers. Four of them (FGFR3, PTEN, MTOR and BRAF) have been specifically associated with testicular tumors”*.

5) The authors state in the discussion that set 1 genes should have a substantial disease effect and moderate CES effect and that the Nanoseq sperm sequencing could be underpowered to identify CES for such genes. But how do the authors explain the CES drivers that were detected by Nanoseq sequencing, but not the author’s method?

We appreciate the reviewer’s question and have addressed it, along with related concerns above, more explicitly in the Main Text. In addition to showing the overlap, we now explicitly

report in the Main Text that 17 out of 40 genes in our CES list are nominally significant in the NanoSeq dataset, supporting the relevance of our findings.

- The NanoSeq data, while offering a powerful approach to identify CES drivers, currently has some limitations that influence overlap. Notably, non-uniform coverage across the genome, which can reduce sensitivity for detecting CES drivers in certain genes.
- The number of observed mutations per gene in the NanoSeq dataset is generally low, which makes it challenging to achieve robust rankings of genes in any statistical setting. We note that our method also generates ranked lists of candidate CES genes based on discrete mutation counts, which should lead to incomplete overlap even if underlying CES effects are identical. To investigate this, we resampled LoF mutations from the NanoSeq dataset and estimated the overlap among $FDR < 0.2$ genes between two independent iterations of the resampling. Based on the results of this analysis, we hypothesize that the effect of inconsistent sorting on finite counts (also called the winner's curse) is the main reason for the incomplete ($\sim \frac{1}{3}$ on average) overlap between sampling replicates from NanoSeq (see Figure below).
- Importantly, our findings are supported by independent lines of evidence beyond the NanoSeq comparison: the signal present in the control cohort as well as cohorts ascertained by conditions other than ASD/NDD and enrichment in cancer driver and CES-relevant pathway. Together, these lines of evidence provide the strong support of our CES candidates.

Distribution for the fraction of FDR-significant genes in the overlap between two samples of mutations from the NanoSeq dataset. Red dashed line corresponds to overlap between FDR-significant genes from Neville et al, and LoF sets 1 and 2.

We also clarified this issue in the Main Text:

“Partial overlap between the lists could be attributed to multiple factors. First of all, datasets obtained by NanoSeq, have variable coverage among the genes, and indeed our CES candidates that are not significant in Neville et al, have lower coverage (Extended Data Figure 6). Second, for GoF-1 and LoF-1 the two approaches have differential statistical power. For the human genetics approach, the power is enhanced by the effect of CES drivers on NDD, whereas the power of the NanoSeq experiment is not boosted by the ascertainment. Finally, the power of both approaches is affected by sampling variance in the low mutation count data.

To demonstrate that the overlap is limited by power, we show that the residual effect exists in genes outside of the overlap. We calculated the observed-to-expected ratio for LoF mutations in the NanoSeq dataset for genes that did not attain genome-wide significance, but are included in our LoF sets (Fig 4b). We also measured the observed-to-expected ratio for genes outside the overlap using de novo variants in control trios (Fig 4b). In both cases, we observe significant enrichment of LoF mutations ($p < 0.01$, Poisson test).”

We would like to use this opportunity to thank the three anonymous reviewers once again for their insightful and constructive feedback, which has helped us substantially improve the quality of our manuscript. Below, please find a point-by-point response to the minor comments raised by the reviewers, as well as a response to the editorial checklist.

Referees' comments

Referee #1 (Remarks to the Author):

The authors have answered satisfactorily to all my comments. Importantly, they have incorporated these answers together with those of the other reviewers into a much-improved manuscript. It is now clear to me exactly how each analysis was performed and on which data. And the findings nicely complement those found by the nanoseq sequencing paper. I still find some of the arguments convoluted but that may just reflect the complexity of an analysis based on an ascertained set of trios and I do not have more suggestions for improving this (but I would appreciate if the authors have some since this could increase the readership of the paper).

My single concrete suggestion is that the authors could consider more in the discussion the evolutionary consequences of the process, like how much selection is needed for CES not to fix gain-of-function mutations, and an estimated total genetic load of the process.

We agree that the questions related to evolutionary consequences of CES are interesting. In response to this comment, we have included the following sentences in the revised Discussion section:

“From the human genetics perspective, CES acts as a factor inflating mutation rate at certain positions. Inflation of mutation rate leads to increase in population frequency of CES drivers. However, because the putative CES drivers reported here are kept at low frequency in the population, we expect that they are subject to strong negative selection even in cases with no obvious disease association.”

It could also be interesting to look up the prevalence of the identified CES mutations as a function of paternal age in the trios (maternal age as well), since CES should be exponentially increasing with age

Interestingly, the prevalence of CES-related conditions such as Apert syndrome and achondroplasia stratified by the age of the father was studied by epidemiologists in the 60-s and in the 70-s. The authors of the original studies really did observe an exponential dependency on the paternal age. In response to this comment, we have included the following in the revised Discussion section:

“Remarkably, prevalence of CES-related diseases such as Apert syndrome and achondroplasia scales exponentially with paternal age. Therefore, it is expected that prevalence of any CES-related condition would similarly show the exponential dependency on the age of the father.”

Referee #2 (Remarks to the Author):

The authors have addressed the major concerns from the initial review, substantially enhancing the manuscript. All primary issues have been resolved, and the statistical tests employed appear appropriate, with the meaning of error bars in figures now clearly defined.

My remaining minor questions/suggestions pertain to wording and figure presentation:

1. To facilitate reader understanding, the meaning of variables V (presence of genetic variant) and D (ascertainment of disease) should be clearly indicated for Equation 1, before discussing other quantities like P(V) and P(D).

We have included the following sentence in the introduction of Eq. 1:

“Denoting V the presence of a de novo variant and D – presence of the disease, we have...”

2. The term "cancerogenesis" on page 5 is less common; "carcinogenesis" is the preferred phrase.

The term "cancerogenesis" has been changed to "carcinogenesis" throughout the manuscript.

3. In the last section of the discussion, "lethality" implies a dichotomous outcome and frequently requires the qualifier "partial." "Embryonic/prenatal deleteriousness" might be a more suitable substitution for "partial embryonic lethality."

The term “lethality” has been changed to “embryonic deleteriousness” throughout.

4. Figures 1d and 4b are too small, hindering the readability of details, including error bars.

Figures 1d and 4b have been restyled to enhance readability.

5. Figs 3b and 4b: the numbers below the bars are confusing. Variant classes should be directly below the bars, perhaps with the number of variants in parentheses beside the classes.

The numbers below the bars on figures 3b and 4b have been removed; variant classes are now placed near the bars: directly either below (in case of vertical bars on Fig. 3b) or to the left (in case of horizontal bars on Fig. 4b)

6. Fig 4d: the in-figure legend ("* significance in NDD") appears inconsistent with the caption's description ("genes with missenses significant in NDD are highlighted with blue shading").

The caption is now consistent with the in-figure legend.

7. Extended data figure 3: The y-axis label (LOELF) is inconsistent with the legend "Observed/Expected Upper bound Fraction."

The legend has been corrected

8. Extended data figure 7: The label of the in-figure color scale legend seems incorrect. Should it be "power" instead of "(1-power)"? Power is expected to increase with mutation rate inflation factor κ and the number of variants, which is the opposite of what the figure currently implies.

The label on Extended Data Fig. 7, " $1-\beta$ ", denotes statistical power (one minus the Type II error rate, β), which is the standard frequentist definition. We have now clarified this in the figure caption.

On a separate point, what does the current sample size lie in these plots? For $\kappa > 20$ (which is reasonable given Fig 4b), isn't there reasonable power at significance level of 0.01 with as few as 20 mutations?

Accurate phasing of *de novo* variants in probands is essential for overtransmission analyses, but most of the current trio datasets were produced with short-read sequencing, which severely limits phasing: in practice only ~10% of samples can be phased with acceptable quality. Thus, although the raw counts in the current datasets (up to ~100 LoF mutations per gene and ~30 GoF mutations per site) appear sufficient for good power even when κ is modest, the phaseable sample size is much smaller rendering these datasets not usable for well-powered overtransmission assessment.

Referee #3 (Remarks to the Author):

Thank you very much to the authors for responding to my comments and for their comprehensive revisions. I believe the manuscript is now substantially improved. Several previously unclear sections have been rewritten with greater clarity and are supported by helpful additional explanations. The explanation for their rationale for using missense variants as a proxy for GoF mutations was particularly helpful. The revised figures are also much improved, especially the replacement of Fig 3a with a more readable table (Fig 1e). I particularly note the more detailed discussion comparing the findings with Nanoseq, which I think strengthens the interpretation of the results.

I am happy that the author's have addressed all my comments/ concerns.

We thank the reviewer for their suggestions and insightful comments.